# Decoupling Variable and Temporal Dependencies: A Novel Approach for Multivariate Time Series Forecasting

## Abstract

In multivariate time series forecasting using the Transformer architecture, capturing temporal dependencies and modeling inter-variable relationships are crucial for improving performance. However, overemphasizing temporal dependencies can destabilize the model, increasing its sensitivity to noise, overfitting, and weakening its ability to capture inter-variable relationships. We propose a new approach called the Temporal-Variable Decoupling Network (TVDN) to address this challenge. This method decouples the modeling of variable dependencies from temporal dependencies and further separates temporal dependencies into historical and predictive sequence dependencies, allowing for a more effective capture of both. Specifically, the simultaneous learning of time-related and variable-related patterns can lead to harmful interference between the two. TVDN first extracts variable dependencies from historical data through a permutation-invariant model and then captures temporal dependencies using a permutation-equivariant model. By decoupling variable and temporal dependencies and historical and predictive sequence dependencies, this approach minimizes interference and allows for complementary extraction of both. Our method provides a concise and innovative approach to enhancing the utilization of temporal features. Experiments on multiple real-world datasets demonstrate that TVDN achieves state-of-the-art (SOTA) performance. The code is available at the repository `https://anonymous.4open.science/r/TVDN-366F`

## 1 Introduction

As artificial intelligence technologies continue to advance, the role of time series forecasting in critical sectors such as energy management(Gao et al., 2023a), meteorology(Meenal et al., 2022), finance(Lopez-Lira & Tang, 2023), and sensor networks(Mejia et al., 2020) has become increasingly important. Long-term Time Series Forecasting (LTSF), involving projections far into the future, is crucial for strategic planning and provides significant reference value.

The limitations of traditional statistical techniques in handling complex time series forecasting tasks have sparked increasing interest among data scientists in applying deep learning methodologies for forecasting. Over years of evolution and competitive advancements, the Time-Series Forecasting Transformer (TSFT), renowned for its superior sequence modeling abilities and scalability, has become widely adopted for long-term time series forecasting.

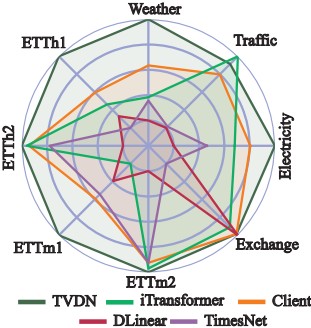

Figure 1: The mean squared error (MSE) of TVDN on various real-world datasets compared with other SOTA methods.

Nonetheless, TSFT models has faced skepticism from researchers(Zeng et al., 2023). Previous studies (Zeng et al., 2023; Gao et al., 2023b) have shown that TSFT's effectiveness remains the same, mainly even when parts of the historical

sequence are masked, leading to doubts about its ability to extract significant information from these sequences.

**Variable dependencies capture** Multivariate time series often show both instantaneous(Gersch, 1985; Koutlis et al., 2019) and lagged effects(Lin et al., 2023), such as transient correlations between heart rate and blood pressure or gradual temperature impacts on plant growth. Specific TSFT models employing cross-variable transformers have made significant progress in long-term forecasting (Liu et al., 2024; Gao et al., 2023b; Zhang & Yan, 2022). These models notably enhance performance, especially in datasets characterized by multivariable interdependencies. Liu et al. (2024); Zeng et al. (2023) think that feed-forward networks (FFN) favor extracting the series representations.

[hsgf]

**Temporal dependencies capture** However, some linear models and Cross-Variable Transformers do not extract accurate temporal dependencies because they essentially map historical series as unordered sets to predicted series experiment 4.3. The reason for their better performance may be that in some tasks, the time dependence of the historical sequence does not contribute much to the prediction of the target sequence. To address the deficiencies of permutation-invariant models, we focus on temporal features, dividing them into the temporal dependencies of the **historical sequences** and the temporal dependencies of the **prediction sequences**.

[tVnS]

**Split Variable Dependencies Learning and Temporal Learning** The vanilla Transformer model divides sequences along the temporal dimension. However, this approach fails to focus on learning the correct patterns, resulting in performance comparable to or even worse than simple linear baselines Zeng et al. (2023). In contrast, cross-variate Transformer models adopt a variable-oriented perspective, splitting sequences along the variable dimension, which significantly improves prediction performance Liu et al. (2024); Gao et al. (2023b). Crossformer (Zhang & Yan, 2022) attempts to capture temporal and variable dependencies simultaneously but still shows room for improvement in prediction accuracy. Our experiments observed that learning both patterns simultaneously leads to performance degradation. Supporting studies have also demonstrated that cross-temporal self-attention can result in bad local minima and make it harder to converge to true solutions. To address this, optimization techniques have been proposed to guide the model toward a better gradient direction Ilbert et al. (2024). Inspired by these findings, we first leverage cross-variate learning to obtain a better initialization point, followed by cross-temporal learning to guide the model toward its true solution.

[hsgf]

In conclusion, based on the analysis above, we introduce a dual-phase deep learning network architecture. The initial phase, the Cross-Variable Encoder (CVE), aims to identify inter-variable dependencies, effectively extracting information from historical sequences. Once the CVE stabilizes, the second phase shifts to temporal dependency learning. In this phase, the Cross-Temporal Encoder (CTE) combines the original input with the output from the CVE, focusing on learning cross-temporal dependencies. This approach addresses the limitations of temporal dependency learning inherent in the first phase's cross-variable feature learning and clarifies the temporal relationships within predictive sequences.

[hsgf]

By segregating cross-variable and cross-temporal learning, our model significantly reduces the risk of overfitting and enhances the potential to discover better global solutions. Our experimental results demonstrate that the proposed TVDN (Temporal-Variable Decoupling Network) achieves state-of-the-art (SOTA) performance on real-world forecasting benchmarks, as illustrated in Figure 1. Our contributions can be summarized in three key aspects:

- This study introduces the Temporal-Variable Decoupling Network (TVDN), which combines permutation-invariant and permutation-equivariant models to decouple variable and temporal dependencies, reducing interference between them and improving temporal feature utilization.

- This study decouples learning into three sub-modes: variable dependency, historical sequence temporal learning, and predicted sequence temporal learning, then integrates them to maximize effectiveness and overcome the limitations of feature extraction in permutation-invariant and permutation-equivariant models.

- TVDN significantly improves multivariate time series forecasting accuracy with minimal overhead, achieving comprehensive SOTA performance on real-world benchmarks. It effectively captures both variable and temporal dependencies. Our analysis of the two-phase architecture highlights its rationale and effectiveness, offering a novel framework for developing more interpretable and accurate forecasting methods.

## 2 RELATED WORK

Traditional time series forecasting methods such as ARIMA(Anderson, 1976), Holt-Winters(Hyndman & Athanasopoulos, 2018), and Exponential Smoothing(Brown, 1959) assume that temporal variations follow fixed patterns. However, real-world time series data often contain complexities that these methods fail to capture, limiting their effectiveness in practical applications(Box et al., 2015; Chatfield & Xing, 2019).

To address the shortcomings of classical models, deep learning approaches have been developed for temporal modeling, including TCN, RNN-based, and MLP-based methods. MLP-based models(Challu et al., 2023; Zeng et al., 2023) utilize MLPs along the temporal dimension to encode temporal dependencies into the fixed parameters of the MLP layers. TCN-based methods capture temporal variations using convolutional kernels that slide along the temporal dimension(Wu et al., 2022). RNN-based methods(Lai et al., 2018; Gu et al., 2021) employ a recurrent structure to implicitly capture temporal variations through state transitions over time.

The Transformer model, celebrated for its exceptional performance in diverse domains such as natural language processing, speech recognition, and computer vision, has been adapted for time series forecasting through various variants to enhance its self-attention mechanism(Vaswani et al., 2017). These adaptations primarily focus on learning long-term dependencies using cross-temporal attention mechanisms and optimizing computational efficiency.

LogTrans(Li et al., 2019) introduces a convolutional self-attention layer with a LogSparse design, adept at capturing local information while reducing spatial complexity. Other models, such as Informer(Zhou et al., 2022a) and Autoformer(Wu et al., 2021), innovate by replacing the traditional self-attention mechanism, lowering computational complexity to $O(L \log L)$. Pyraformer(Liu et al., 2021) integrates pyramid attention modules that connect across and within scales, achieving linear complexity.

Further advancements include models like Autoformer, FEDformer(Zhou et al., 2022b), and ETSformer(Woo et al., 2022), which incorporate TSFT with seasonal trend decomposition and signal processing techniques, such as Fourier analysis, within their attention frameworks. This enhances the interpretability of these models and efficiently captures seasonal trends.

To address stability in predictions, especially in non-stationary contexts, some Transformer models incorporate stabilization modules and De-stationary into the standard Transformer framework(Liu et al., 2022; Kim et al., 2021). This helps stabilize predictions while avoiding the pitfalls of excessive stabilization, which can lead to a loss of important data variability.

Recent developments in cross-variable Transformer models show significant promise. Models like iTransformer(Liu et al., 2024) and Client(Gao et al., 2023b) enhance performance in long-term multivariate forecasting by using cross-variable Transformers instead of cross-temporal ones. Additionally, Crossformer(Zhang & Yan, 2022) employs a two-stage attention (TSA) layer to capture dependencies over time and across different dimensional segments of the series. However, there is room for improvement in models like Crossformer regarding their performance on various benchmark datasets.A recent work PatchTST (Nie et al., 2022) studies using a vision transformer type model for long-term forecasting with channel independent design. This work designs an encoder-decoder model utilizing a hierarchy attention mechanism to leverage cross-dimension dependencies.

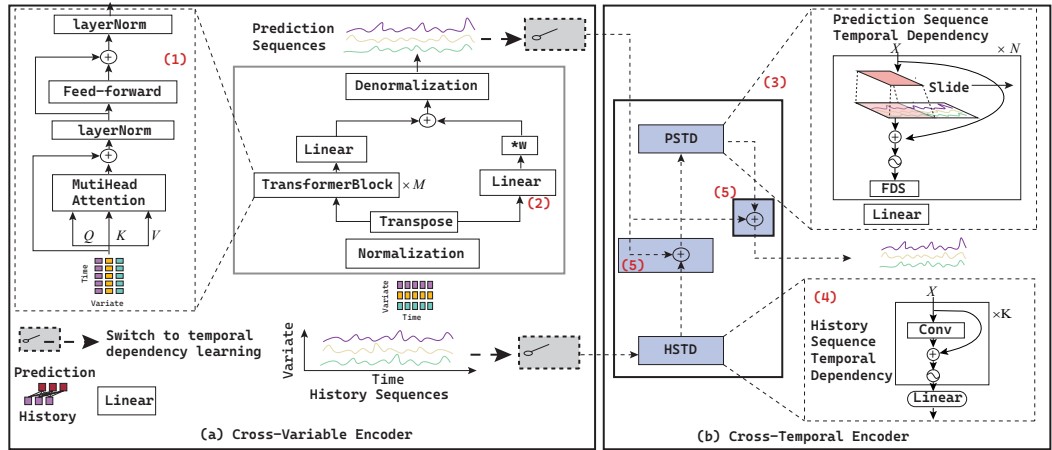

Figure 2: **Overview of the proposed method**. (1) Cross-Variable Transformer. (2) Linear Model (3) Prediction Sequence Temporal Dependency Learning Module. (4) Historical Sequence Temporal Dependency Learning Module (5) Feature Fusion. The TVDN architecture is strategically bifurcated into two key components. On the left, CVE leverages the Cross-Variable Transformer to effectively delineate dependencies among variables. In contrast, on the right, CTE utilizes (3) to capture prediction sequences temporal dependencies and (4) to capture historical sequences temporal dependencies.

## 3 MODEL ARCHITECTURE

The architecture of TVDN is depicted in Figure 2. As previously discussed, we separate the learning of variable dependency from that of temporal dependency. The process begins with variable dependency learning (left), followed by temporal dependency learning(right), which is further divided into two sub-modules: historical sequence dependency and predictive sequence dependency.

### 3.1 CROSS-VARIABLE ENCODER (CVE)

CVE is a permutation-invariant model used for modeling variable dependencies. CVE is based on the Cross-Variable Transformer (Liu et al., 2024; Gao et al., 2023b), which treats the input data as a sequence of variables to capture complex dependencies among them. The hallmark of CVE lies in its novel approach to token partitioning. Unlike traditional methods, CVE segments tokens along the variable dimension, with each token representing different temporal instances of the same variable. This is achieved by transposing the input data. The process is illustrated as follows:

$$\mathbf{V}^0 = \text{Transpose}(\mathbf{X}_{\text{enc}}) \tag{1}$$

$$\mathbf{V}^{(m+1)} = \text{TransformerBlock}(\mathbf{V}^m), \quad m \in \{0, 1, \ldots, M-1\} \tag{2}$$

$$\mathbf{Z}_{\text{CVE}} = \text{Projection}(\mathbf{V}^M) + weight \times \text{Projection}(\mathbf{X}_{\text{enc}}) \tag{3}$$

The operational sequence begins by transposing the input data $\mathbf{X}_{\text{enc}}$ to form $\mathbf{V}^0$, where $\mathbf{V}$ is a matrix containing $D$ embedded tokens, each with a dimension of $S$. $D$ is equal to the number of variables, $S$ is the length of time series, and weight is a learnable parameter. Here, $\mathbf{V}^0 \in \mathbb{R}^{D \times S}$ represents the initial embedded form of the input. The superscript in $\mathbf{V}^{(m+1)}$ indicates the layer index in the progression of transformations.

Each subsequent layer $\mathbf{V}^{(m+1)}$ is generated by applying a *TransformerBlock* to the output of the previous layer $\mathbf{V}^m$. This process is repeated for $m \in \{0, 1, \ldots, M-1\}$. The

*TransformerBlock* typically consists of self-attention mechanisms and a shared feed-forward network (FFN), allowing the variable tokens within $\mathbf{V}$ to interact and be processed independently at each layer. This iterative process enriches the data representation by capturing complex dependencies and patterns.

Finally, the *Projection* operation transforms the output of the last Transformer layer $\mathbf{V}^M$ and the original input data $\mathbf{X}_{\text{enc}}$ into a common space, which is then added with a learnable weight *weight* to obtain the final output $\mathbf{Z}_{\text{CVE}}$, where $\mathbf{Z}_{\text{CVE}} \in \mathbb{R}^{O \times D}$ and $O$ represents the prediction length. This output is then used as input to the next phase of learning, the Cross-Temporal Encoder (CTE). To address the issue of distribution shift, CVE employs a reversible instance normalization (RevIN) module (Kim et al., 2021). This module, characterized by its symmetrical structure, can remove and restore the statistical information of time series instances, thereby enhancing the model's stability during the prediction process.

[tVnS]

CVE channels the extracted features into a projection layer to generate first-stage predictions, deliberately omitting a Transformer decoder. This approach stems from the decoder's inherent assumption of future sequence invisibility, which overlooks the constraining influence of future sequences on historical data. Additionally, the Transformer module within CVE operates predominantly as a feature extractor rather than a sequence generator, given the absence of temporal interrelations among different variables.

## 3.2 Cross-Temporal Encoder (CTE)

The CTE plays a crucial role in modeling the temporal dependencies. CTE divides time series dependence into two parts: historical sequences dependence and predictive sequences dependence. It processes inputs that include the outputs of the original historical sequences combined with the results from the CVE. This combination of data allows the CTE to effectively capture the temporal dependencies of the history sequences and prediction sequences, overcoming the CVE stage's limitations in recognizing dynamic temporal characteristics.

The output of the CTE is then combined with the output of the CVE through an additive fusion process to optimize the residual between the CTE and the predictive sequence. The CTE is simply expressed as:

$$\mathbf{Z}_h = \text{HSTDBlock}(\mathbf{V}^0) \tag{4}$$

$$\mathbf{T}^0 = \mathbf{Z}_{\text{h}} \oplus \mathbf{Z}_{\text{CVE}} \tag{5}$$

$$\mathbf{T}^{n+1} = \text{FDS}(\text{PSTDBlock}(\mathbf{T}^n)), \quad \text{for } n \in \{0, 1, \ldots, N-1\} \tag{6}$$

$$\mathbf{Y} = \mathbf{Z}_{\text{CVE}} \oplus \text{Projection}(\mathbf{T}^N) \tag{7}$$

where $\mathbf{T}^0$ denotes the initial input state, formed by the addition of $\mathbf{Z}_{\text{h}}$ and $\mathbf{Z}_{\text{CVE}}$, where $\mathbf{T}^0$ resides in the space $\mathbb{R}^{O \times D}$. This signifies that $\mathbf{T}^0$ contains $O$ embedded tokens, each of dimension $D$, capturing the combined information from the projected target sequence and the output of CVE. $n$ indicates the layer index in the sequence of transformations, iterating from 0 to $N-1$. FDS and the CrossTimeBlock interactively refine the temporal features in each layer. Finally, the cumulative output of this sequential operation, $\mathbf{T}^N$, is combined with the CVE's output.

**Prediction Sequence Temporal Dependency (PSTD)** The role of PSTD is to model the time dependence of prediction sequences. The PSTD block consists of a convolutional layer and employs a concatenation operation to ensure that no information is lost from the input. To avoid performance degradation and the risk of overfitting due to an excess of features, we employ point-wise convolutions to construct a Feature Down-Sample (FDS) module, which halves the input features.

**Historical Sequence Temporal Dependency (HSTD)** The role of HSTD is to model the time dependence of historical sequences. The HSTD block consists of a convolutional layer and employs a residual connection to ensure that important historical information is retained and to prevent performance degradation as the network deepens.

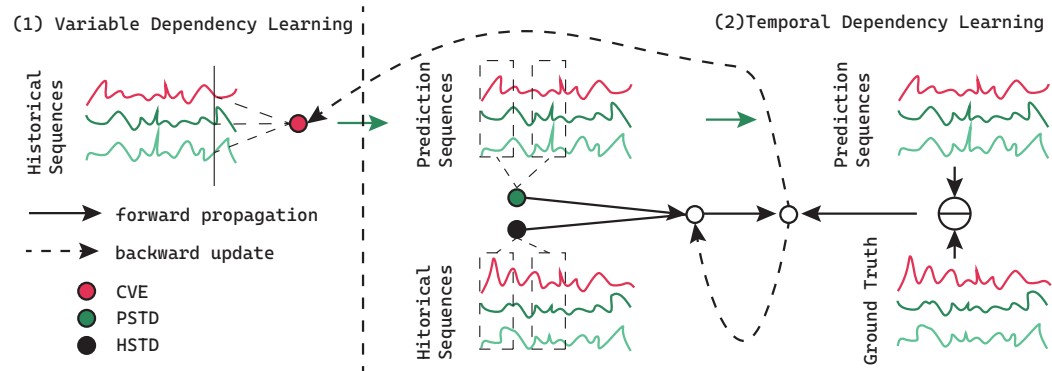

Figure 3: **Overview of the training process**

**Feature Down-Sample (FDS)**. The input data and encoding process generate many redundant features. FDS is used to suppress these redundant features generated during the encoding process while eliminating the performance overhead caused by channel expansion.

### 3.3 TRAINING PROCESS

As shown in Figure3, first, during the variable dependence learning phase, the permutation-invariant CVE completely disregards the temporal dependence of the sequence and only extracts cross-features between variables, generating an initial prediction sequence. At the same time, the CTE remains frozen at this stage. Next, HSTD extracts the temporal features of the historical sequences, while PSTD extracts the temporal features of the prediction sequences. The outputs of HSTD and PSTD are then fused to correct the initial prediction from the variable dependence learning phase (residual fitting). At the same time, the CVE and CTE model parameters are updated through backpropagation.

### 4 EXPERIMENTS

**Datasets** In this study, we evaluate the performance of TVDN using eight popular datasets from various fields, including electricity(Trindade, 2015), traffic(pem), weather(Max-Planck-Institut für Biogeochemie), four ETT (Electricity Transformer Temperature, including ETTh1, ETTh2, ETTm1, and ETTm2)(Zhou et al., 2021), and exchange(Lai et al., 2018).

### 4.1 MAIN RESULTS

**Baselines** We compared the latest TSFT methods(iTransformer(Liu et al., 2024), Client(Gao et al., 2023b), LightTS(Zhang et al., 2022), FEDformer(Zhou et al., 2022b), Aut-oformer(Wu et al., 2021), ETSformer(Woo et al., 2022), (Zhou et al., 2022a), Pyraformer(Liu et al., 2021)), CNN-based TimesNet (Wu et al., 2022), and linear model Dlinear(Zeng et al., 2023).

**Experimental Settings** The look-back window size for all datasets is uniformly set at 96, and the number of training epochs is fixed at 10 for each. We assess the performance using four different prediction lengths $\{96, 192, 336, 720\}$. Following the evaluation procedure used in previous studies, we compute the Mean Squared Error (MSE) and Mean Absolute Error (MAE) for data normalized with z-score normalization.

**Results** The long-term sequence forecasting results are presented in Table 1, Table 3, Table 4 and Figure 9. We maintained consistency in the look-back window and training epochs to ensure the most equitable comparison.

Both iTransformer and Client use a cross-variable Transformer architecture, ranking just below TVDN. It shows that models ignoring temporal ordering can capture cross-variable relationships more effectively, partly supporting the hypothesis that learning temporal de-

pendencies may interfere with variable dependencies. DLinear excelled on the Exchange dataset, which has fewer variables, indicating its strength in forecasting scenarios focused on single variables. FEDformer leverages frequency domain analysis and performed well on the ETTh1 dataset, highlighting the importance of frequency domain features. TimesNet, which transforms time series into two-dimensional tensors to capture both intra-periodic and inter-periodic patterns, showed strong performance on ETTh1 and ETTm2, aligning with the emphasis on periodicity and locality in sequences.

TVDN surpasses all SOTA models, achieving the best performance on several popular datasets. Overall, it achieved first place in 70 (Second best model is 12) categories, and it leads other advanced models by a significant margin in both the average and median numbers of first places in MSE and MAE.

TVDN, through its CVE, thoroughly mines variable dependencies from historical sequences and, through its CTE, fully learns the temporal dependencies of the prediction and historical sequence. (1) By separating and training cross-variable and cross-time learning, we avoided mixing the two learning modes, enhancing the prediction results. (2) The motivation for incorporating the temporal dependence of the prediction series into the model is: Based on our experiments F, we identified that the bottleneck of the traditional Transformer model lies in the ineffective utilization of **historical sequence** information. Its primary benefit is learning the temporal dependency patterns of the **prediction sequence**.

Table 1: Multivariate forecasting results with prediction lengths (96, 192, 336, 720). Results are averaged from all prediction lengths. Avg means further averaged by subsets. Me means the mean of the results. The best results and second-best results are highlighted in red and blue, respectively. Full results are listed in Appendix 3

| Models Metric | | TVDN MSE MAE | iTransformer MSE MAE | Client MSE MAE | DLinear MSE MAE | TimesNet MSE MAE | FEDformer MSE MAE | ETSformer MSE MAE | LightTS MSE MAE | Autoformer MSE MAE | Pyraformer MSE MAE | Informer MSE MAE |
|---|---|---|---|---|---|---|---|---|---|---|---|---|
| Electricity | Avg | **0.158 0.256** | 0.178 0.270 | 0.171 0.264 | 0.212 0.300 | 0.192 0.295 | 0.214 0.327 | 0.208 0.323 | 0.229 0.329 | 0.227 0.338 | 0.379 0.445 | 0.311 0.397 |
| | Me | **0.158 0.257** | 0.170 0.261 | 0.167 0.261 | 0.203 0.293 | 0.191 0.295 | 0.208 0.322 | 0.206 0.322 | 0.222 0.325 | 0.227 0.336 | 0.377 0.444 | 0.298 0.390 |
| Traffic | Avg | 0.433 **0.265** | **0.428** 0.282 | 0.465 0.304 | 0.625 0.383 | 0.620 0.336 | 0.610 0.376 | 0.621 0.396 | 0.622 0.392 | 0.628 0.379 | 0.878 0.469 | 0.764 0.416 |
| | Me | 0.432 **0.265** | **0.425** 0.280 | 0.462 0.302 | 0.625 0.384 | 0.623 0.336 | 0.613 0.378 | 0.622 0.396 | 0.614 0.389 | 0.619 0.385 | 0.875 0.469 | 0.748 0.406 |
| Weather | Avg | **0.234** 0.276 | 0.258 0.279 | 0.249 **0.275** | 0.265 0.317 | 0.259 0.287 | 0.309 0.360 | 0.271 0.334 | 0.261 0.312 | 0.338 0.382 | 0.946 0.717 | 0.634 0.548 |
| | Me | **0.230** 0.277 | 0.250 0.275 | 0.243 **0.274** | 0.260 0.316 | 0.250 0.284 | 0.308 0.358 | 0.268 0.333 | 0.255 0.311 | 0.333 0.381 | 0.872 0.689 | 0.588 0.534 |
| ETTh1 | Avg | 0.445 **0.437** | 0.454 0.447 | 0.452 0.445 | 0.456 0.452 | 0.458 0.450 | **0.440** 0.460 | 0.542 0.510 | 0.491 0.479 | 0.496 0.487 | 0.827 0.703 | 1.040 0.795 |
| | Me | 0.458 **0.441** | 0.464 0.447 | 0.464 0.446 | 0.459 0.446 | 0.464 0.449 | **0.440** 0.457 | 0.550 0.513 | 0.497 0.475 | 0.507 0.489 | 0.841 0.710 | 1.058 0.801 |
| ETTh2 | Avg | **0.373 0.402** | 0.383 0.407 | 0.386 0.411 | 0.559 0.515 | 0.414 0.427 | 0.437 0.449 | 0.439 0.452 | 0.602 0.543 | 0.450 0.459 | 0.826 0.703 | 4.431 1.729 |
| | Me | **0.386 0.409** | 0.404 0.416 | 0.403 0.423 | 0.536 0.509 | 0.427 0.433 | 0.446 0.457 | 0.458 0.459 | 0.573 0.532 | 0.469 0.469 | 0.848 0.715 | 4.238 1.730 |
| ETTm1 | Avg | **0.388 0.395** | 0.407 0.410 | 0.399 0.401 | 0.403 0.407 | 0.400 0.406 | 0.448 0.452 | 0.429 0.425 | 0.435 0.437 | 0.588 0.517 | 0.691 0.607 | 0.961 0.734 |
| | Me | **0.380 0.393** | 0.402 0.406 | 0.391 0.397 | 0.397 0.401 | 0.392 0.399 | 0.436 0.450 | 0.422 0.419 | 0.419 0.423 | 0.587 0.517 | 0.656 0.596 | 0.981 0.746 |
| ETTm2 | Avg | **0.285 0.327** | 0.288 0.332 | 0.291 0.330 | 0.350 0.401 | 0.291 0.333 | 0.305 0.349 | 0.293 0.342 | 0.409 0.436 | 0.327 0.371 | 1.498 0.869 | 1.410 0.810 |
| | Me | **0.276 0.323** | 0.281 0.329 | 0.283 0.326 | 0.327 0.395 | 0.285 0.330 | 0.297 0.347 | 0.284 0.338 | 0.377 0.424 | 0.310 0.356 | 0.966 0.759 | 0.948 0.725 |
| Exchange | Avg | **0.345** 0.405 | 0.360 **0.403** | 0.355 **0.403** | 0.354 0.414 | 0.416 0.443 | 0.519 0.500 | 0.410 0.427 | 0.385 0.447 | 0.613 0.539 | 1.913 1.159 | 1.550 0.998 |
| | Me | 0.258 0.372 | 0.254 **0.358** | 0.253 **0.358** | **0.245** 0.371 | 0.297 0.396 | 0.366 0.440 | 0.265 0.366 | 0.296 0.413 | 0.405 0.447 | 1.909 1.162 | 1.438 0.966 |

## 4.2 Influence Of Splitting Variable And Temporal Learning

In this section, we conducted ablation experiments on three datasets to verify the necessity and effectiveness of treating the input sequence as a variable and then switching to a time sequence in TVDN. The experiment results are shown in Figure 16, Figure 5 and Table 4. Significantly decreases without CTE. This means that CTE fully complements the learning of temporal dependent features.

**Decoupling effect** As shown in Table 4, the model's performance deteriorates when trained by CVE and CTE. This suggests that simultaneous cross-variable and cross-temporal learning can cause mutual interference. The process of temporal dependency learning is prone to transmitting the effects of overfitting to variable dependency learning. However, performing variable dependency learning first and switching to temporal dependency learning can effectively avoid these issues. This approach allows the model to gradually adapt to different aspects of the data rather than trying to fit all complex relationships simultaneously. The method of decoupling temporal features from variable features achieved 15 first-place counts

in MSE and 14 first-place counts in MAE, demonstrating a significant advantage over the CVE model, which only captures variable dependencies and non-decoupling methods.

Figure 4: Comparison of joint (TVDN-mix) and decoupled (TVDN-split) training strategies for CVE and CTE modules.

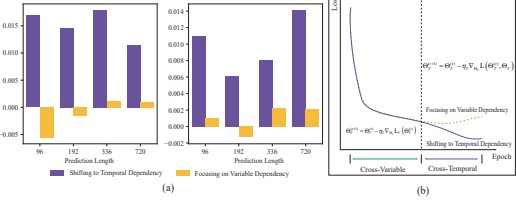

| Method | | TVDN-mix | | TVDN-split | | CVE | |
|---|---|---|---|---|---|---|---|
| Metric | | MSE | MAE | MSE | MAE | MSE | MAE |
| ECL | 96 | 0.140 | 0.236 | **0.132** | **0.226** | 0.142 | 0.238 |
| | 192 | 0.161 | 0.254 | **0.153** | **0.250** | 0.160 | 0.252 |
| | 336 | 0.175 | 0.269 | **0.164** | **0.264** | 0.173 | 0.267 |
| | 720 | 0.212 | 0.300 | **0.186** | **0.284** | 0.204 | 0.296 |
| | AVG | 0.172 | 0.265 | **0.158** | **0.256** | 0.170 | 0.263 |
| Traffic | 96 | 0.434 | 0.291 | **0.401** | **0.248** | 0.439 | 0.294 |
| | 192 | 0.453 | 0.297 | **0.427** | **0.259** | 0.455 | 0.299 |
| | 336 | 0.470 | 0.306 | **0.438** | **0.271** | 0.468 | 0.304 |
| | 720 | 0.503 | 0.322 | **0.469** | **0.285** | 0.499 | 0.321 |
| | AVG | 0.465 | 0.304 | **0.433** | **0.265** | 0.465 | 0.305 |
| Weather | 96 | 0.166 | 0.212 | **0.152** | **0.202** | 0.165 | 0.210 |
| | 192 | 0.214 | 0.254 | **0.200** | **0.250** | 0.212 | 0.252 |
| | 336 | 0.272 | **0.294** | 0.261 | 0.305 | 0.270 | **0.294** |
| | 720 | 0.350 | 0.346 | **0.325** | **0.349** | 0.354 | 0.349 |
| | AVG | 0.250 | 0.276 | **0.234** | **0.276** | 0.250 | 0.276 |
| $1^{st}$ count | | 0 | 1 | **15** | **14** | 0 | 1 |

Figure 5: (a) Comparison of MSE reduction on the test Set between shifting to temporal dependency learning and focusing on variable dependency learning. (b) Trend illustration of shifting to temporal dependency and focusing on variable dependency on the validation and test sets. This trend is observed across all the datasets we tested.

The analysis in Figure 16 illustrates how shifting from cross-variable learning to temporal dependency learning approaches improves the model's ability to capture both amplitude and trend characteristics. This phenomenon is observed across multiple datasets, suggesting the robustness of the proposed method. The results highlight the significance of designing a learning strategy that aligns with the temporal and variable dependencies in the data.

**Switching from variable learning to temporal learning** As shown in Figure 5, continuing to learn dependencies among variables results in a minimal decrease in MSE and can even lead to an increase in MSE, making overfitting more likely. However, after switching to temporal dependency learning, the MSE exhibits a secondary decline trend, significantly reducing MSE. As shown in Figure 16, the separated training method has significant advantages in predicting the sequence's amplitude and overall trend. These indicate that TVDN can help the optimization algorithm avoid suboptimal local minima. By shifting the focus of learning, the model may explore a broader parameter space, thereby finding a better global solution.

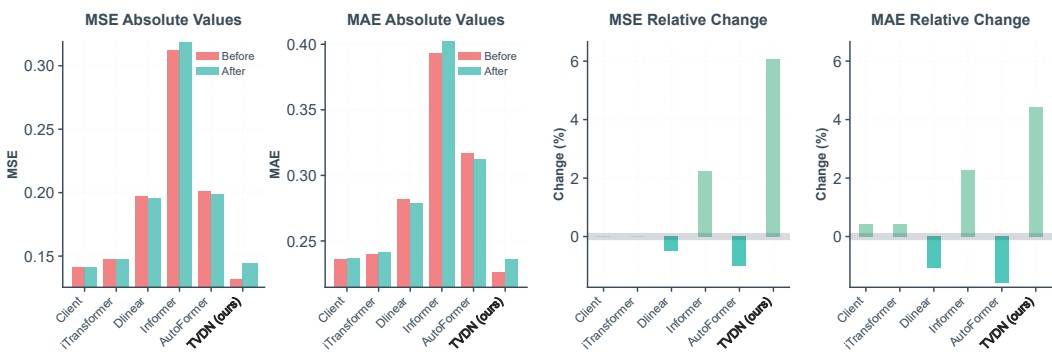

Figure 6: The relative change in MSE and MAE after randomly shuffling historical sequences (Electricity dataset, sequence length=96). TVDN shows the highest increase in errors, indicating it benefits the most from temporal features, while maintaining the lowest absolute MSE/MAE values, suggesting temporal disruption does not impair its cross-variable learning capability.

### 4.3 INFLUENCE OF TEMPORAL FEATURES

To investigate the contribution of temporal features, we designed an experiment on the ECL dataset with input length 96 and prediction length 96, where the time series order was randomized entirely, removing all temporal information. We then observed the change

in performance metrics before and after the randomization to assess the model's reliance on temporal features. The more MSE and MSE grow, the more capable the model is of extracting and utilizing temporal information.

**Results** The results are shown in the Figure 6. After randomly adjusting the time order, the TVDN model has the largest rate of performance degradation, which indicates its strong dependence on the time sequence, and its full extraction of the time sequence features, when the time sequence features are artificially eliminated, he model has the largest performance degradation.

The permutation-invariant Cross-Variable Transformer and Dlinear models remained unaffected, indicating they did not rely on temporal features from historical sequences. In contrast, other permutation-equivariant models (Informer and Autoformer) showed minimal changes in MSE, suggesting a lesser dependence on temporal features. While they did utilize some temporal information, it was insufficient for optimal performance.

### 4.4 MODEL ANALYSIS

**Robustness** As show in Fig. 7 and Appendix E, the robustness of TVDN is tested on the ECL dataset with different levels of Gaussian noise and missing rate levels. The performance of TVDN decreases as the noise level increases, but the decrease is small and stable, which indicates that it is more resistant to noise and has good performance at different noise levels.

[Jesy]

**Efficiency** Figure 8 and Table 8 demonstrate that TVDN achieves superior prediction performance with high efficiency. It requires only 0.46G FLOPs, 1.44M parameters, and 50.25MB peak memory, significantly reducing computational and memory overhead compared to models like iTransformer and TimesNet. TVDN's inference speed is comparable to lightweight models like Client and much faster than TimesNet. Although DLinear has lower costs, it performs worse in prediction accuracy. These results confirm TVDN's balance between efficiency and accuracy.

[Jesy]

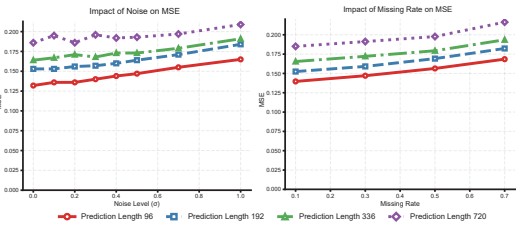

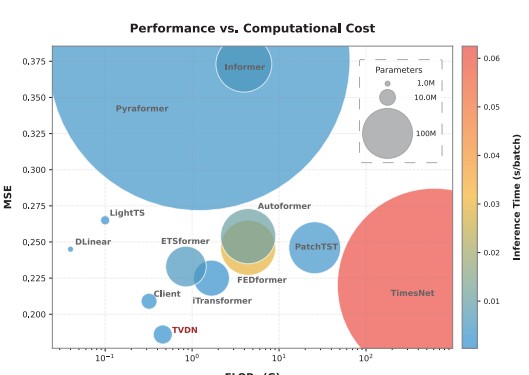

Figure 7: The robustness tests of models on the ECL dataset include performance under varying levels of Gaussian noise (left) and different missing rate levels (right). The Gaussian noise level $\sigma$ indicates that 68% of the noise falls within $\pm\sigma$ of the standardized data. The missing ratio $m$ indicates that $(m\times100)\%$ of the input data points are randomly masked as missing values (set to zero).

Figure 8: Performance and computational cost comparison among different models. The x-axis represents computational complexity in FLOPs, and the y-axis shows MSE. The size of each bubble indicates the number of model parameters, while the color indicates inference time per batch (s/batch) ranging from low (blue) to high (red). TVDN achieves competitive performance with moderate computational cost and relatively small model size.

### 5 CONCLUSION

This paper introduces a method called TVDN, which decouples variable learning from temporal dependency learning and models temporal features through historical and prediction sequence dependency. TVDN effectively minimizes interference, reduces the risk of overfitting, and enables broader parameter space exploration. Experimental results demonstrate that TVDN addresses the limitations of permutation-invariant models in capturing dynamic temporal dependencies and outperforms permutation-equivariant models in efficiently capturing temporal features. TVDN achieves SOTA performance across various real-world datasets.

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

## A  DETAILS OF EXPERIMENTS

### A.1  DATASETS

Table 2: Detailed dataset descriptions. *Dimension* denotes the variate number of each dataset. *Dataset Size* denotes the total number of time points in (Train, Validation, Test) split respectively. *Prediction Length* denotes the future time points to be predicted and four prediction settings are included in each dataset. *Frequency* denotes the sampling interval of time points.

| Dataset | Dimension | Prediction Length | Dataset Size | Frequency |
|---------|-----------|-------------------|--------------|-----------|
| ETTh1, ETTh2 | 7 | {96, 192, 336, 720} | (8545, 2881, 2881) | Hourly |
| ETTm1, ETTm2 | 7 | {96, 192, 336, 720} | (34465, 11521, 11521) | 15min |
| Exchange | 8 | {96, 192, 336, 720} | (5120, 665, 1422) | Daily |
| Weather | 21 | {96, 192, 336, 720} | (36792, 5271, 10540) | 10min |
| ECL | 321 | {96, 192, 336, 720} | (18317, 2633, 5261) | Hourly |
| Traffic | 862 | {96, 192, 336, 720} | (12185, 1757, 3509) | Hourly |

We performed comprehensive evaluations across seven widely adopted time series datasets. In line with previous studies Wu et al. (2022), we split the datasets chronologically to form the training, validation, and testing subsets. Specifically, the ETT dataset was divided with a 6:2:2 ratio, while the remaining datasets employed a 7:1:2 ratio. Below is a summary of the datasets:

- **ETT (Electricity Transformer Temperature)**: This dataset consists of data from electricity transformers located in two regions of China, covering the period from July 2016 to July 2018. It provides two levels of temporal resolution: ETTh (hourly) and ETTm (every 15 minutes). The dataset includes measurements of oil temperature and six external load features.
- **Weather**: The Weather dataset offers meteorological data collected every 10 minutes in Germany throughout 2020. The dataset includes 21 variables, such as air temperature, visibility, and others.
- **Electricity**: This dataset contains hourly electricity usage data from 321 households, recorded between 2012 and 2014. The electricity consumption is measured in kilowatt-hours (kWh), and the data is available from the UCL Machine Learning Repository.
- **Traffic**: The Traffic dataset records hourly road occupancy rates from 862 real-time sensors on highways in the San Francisco Bay Area. The data spans the years 2015 to 2016.

The ETT dataset can be accessed at `https://github.com/zhouhaoyi/Informer2020`, while the other datasets are available at `https://github.com/thuml/Autoformer`. Table 7 provides detailed dataset statistics, including time steps, variables, temporal resolution, and the top five dominant periods.

### A.2  BASELINES

iTransformer (Liu et al., 2024) introduces an innovative inversion of the traditional Transformer architecture for time series forecasting. Instead of embedding time steps, iTransformer treats each variable as an independent token, using self-attention to capture multivariate correlations. This design allows the model to better generalize across different time series, providing improved accuracy and interpretability. The source code can be accessed at `https://github.com/thuml/iTransformer`

FITS (Xu et al., 2024) is a lightweight time series analysis model. It transforms input sequences into the frequency domain, applies a low-pass filter to remove high-frequency

noise, and utilizes a complex-valued linear layer for interpolation, learning amplitude scaling and phase shifting. The processed data is then converted back to the time domain via inverse Fourier transform. This approach enables FITS to excel in tasks like time series forecasting and anomaly detection, with a model size of approximately 10,000 parameters, making it suitable for deployment on resource-constrained edge devices. The source code is available at `https://github.com/VEWOXIC/FITS`.

WITRAN (Jia et al., 2024) introduces a novel framework that captures both long- and short-term patterns through bi-granular information transmission. It employs a Horizontal Vertical Gated Selective Unit (HVGSU) to model global and local correlations and incorporates a Recurrent Acceleration Network (RAN) to enhance computational efficiency. The source code is available at `https://github.com/Water2sea/WITRAN`.

[hsgf]

Client is a model designed for capturing cross-variable dependencies, integrating trend detection and a Reversible Instance Normalization (RevIN) module. The source code is available at `https://github.com/daxin007/Client`

DLinear (Zeng et al., 2023), a simple one-layer linear model, challenges the dominance of Transformer-based models in long-term time series forecasting by demonstrating superior performance across multiple datasets. The source code can be accessed at `https://github.com/vivva/DLinear`.

TimesNet (Wu et al., 2022) is a CNN-based model that converts one-dimensional time series into two-dimensional tensors to effectively capture complex temporal dynamics through adaptive multi-periodicity and inception blocks. The source code is accessible at `https://github.com/thuml/TimesNet`.

FEDformer (Zhou et al., 2022b) leverages a Transformer-based architecture that combines seasonal-trend decomposition with frequency enhancement, enabling it to efficiently capture both global temporal trends and intricate patterns. The source code can be found at `https://github.com/MAZiqing/FEDformer`.

ETSformer (Woo et al., 2022), inspired by exponential smoothing, incorporates both trend and seasonal components into a Transformer architecture. This enables ETSformer to accurately model short- and long-term dependencies in time series data. The source code is available at `https://github.com/salesforce/ETSformer`

LightTS (Zhang et al., 2022) is a lightweight Transformer model designed for long-term time series forecasting. It reduces computational complexity while maintaining accuracy, making it ideal for environments with resource constraints. The source code can be accessed at `https://github.com/d-gcc/LightTS`

Autoformer (Wu et al., 2021) employs a decomposition strategy to separate time series into trend and seasonal components. This approach enhances long-term forecasting by focusing on individual components, allowing the model to learn more effectively. The source code is available at `https://github.com/thuml/Autoformer`

Pyraformer (Liu et al., 2021) utilizes a pyramid structure within its Transformer model to capture hierarchical dependencies over different time scales. This design improves the model's ability to handle both local and global temporal patterns. The source code is accessible at `https://github.com/ant-research/Pyraformer`

Informer (Zhou et al., 2021), known for its ProbSparse Attention mechanism, enhances the efficiency and scalability of Transformer models for long-term time series forecasting. This method reduces the computational complexity of handling long sequences, making it a practical solution for large-scale time series data. The source code is available at `https://github.com/zhouhaoyi/Informer2020`

# B EXTENDED NUMERICAL RESULTS OF TVDN IN LONG-TERM FORECASTING WITH 96 INPUT LENGTH

Table 3: The complete results for LTSF. The results of 4 different prediction lengths of different models are listed in the table. The look-back window sizes are set to 96 for all datasets. We also calculate the average (Avg) and median(Me) of the results for the 4 prediction lengths and the number of optimal values obtained by different models.

| | Models | TVDN | | iTransformer 2024 | | Client 2023b | | DLinear 2023 | | TimesNet 2022 | | FEDformer 2022b | | ETSformer 2022 | | LightTS 2022 | | Autoformer 2021 | | Pyraformer 2021 | | Informer 2021 | |
| --- | --- | --- | --- | --- | --- | --- | --- | --- | --- | --- | --- | --- | --- | --- | --- | --- | --- | --- | --- | --- | --- | --- | --- |
| | Metric | MSE | MAE | MSE | MAE | MSE | MAE | MSE | MAE | MSE | MAE | MSE | MAE | MSE | MAE | MSE | MAE | MSE | MAE | MSE | MAE | MSE | MAE |
| Electricity | 96 | 0.132 | 0.226 | 0.148 | 0.240 | 0.141 | 0.236 | 0.197 | 0.282 | 0.168 | 0.272 | 0.193 | 0.308 | 0.187 | 0.304 | 0.207 | 0.307 | 0.201 | 0.317 | 0.386 | 0.449 | 0.274 | 0.368 |
| | 192 | 0.153 | 0.250 | 0.162 | 0.253 | 0.161 | 0.254 | 0.196 | 0.285 | 0.184 | 0.289 | 0.201 | 0.315 | 0.199 | 0.315 | 0.213 | 0.316 | 0.222 | 0.334 | 0.378 | 0.443 | 0.296 | 0.386 |
| | 336 | 0.164 | 0.264 | 0.178 | 0.269 | 0.173 | 0.267 | 0.209 | 0.301 | 0.198 | 0.300 | 0.214 | 0.329 | 0.212 | 0.329 | 0.230 | 0.333 | 0.231 | 0.338 | 0.376 | 0.443 | 0.300 | 0.394 |
| | 720 | 0.186 | 0.284 | 0.225 | 0.317 | 0.209 | 0.299 | 0.245 | 0.333 | 0.220 | 0.320 | 0.246 | 0.355 | 0.233 | 0.245 | 0.265 | 0.360 | 0.254 | 0.361 | 0.376 | 0.445 | 0.373 | 0.439 |
| | Avg | 0.158 | 0.256 | 0.178 | 0.270 | 0.171 | 0.264 | 0.212 | 0.300 | 0.192 | 0.295 | 0.214 | 0.327 | 0.208 | 0.323 | 0.229 | 0.329 | 0.227 | 0.338 | 0.379 | 0.445 | 0.311 | 0.397 |
| | Me | 0.158 | 0.257 | 0.170 | 0.261 | 0.167 | 0.261 | 0.203 | 0.293 | 0.191 | 0.295 | 0.208 | 0.322 | 0.206 | 0.322 | 0.222 | 0.325 | 0.227 | 0.336 | 0.377 | 0.444 | 0.298 | 0.390 |
| Traffic | 96 | 0.401 | 0.248 | 0.395 | 0.268 | 0.438 | 0.292 | 0.650 | 0.396 | 0.593 | 0.321 | 0.587 | 0.366 | 0.607 | 0.392 | 0.615 | 0.391 | 0.613 | 0.388 | 0.867 | 0.468 | 0.719 | 0.391 |
| | 192 | 0.427 | 0.259 | 0.417 | 0.276 | 0.451 | 0.298 | 0.598 | 0.370 | 0.617 | 0.336 | 0.604 | 0.373 | 0.621 | 0.399 | 0.601 | 0.382 | 0.616 | 0.382 | 0.869 | 0.467 | 0.696 | 0.379 |
| | 336 | 0.438 | 0.271 | 0.433 | 0.283 | 0.472 | 0.305 | 0.605 | 0.373 | 0.629 | 0.336 | 0.621 | 0.383 | 0.622 | 0.399 | 0.613 | 0.386 | 0.622 | 0.337 | 0.881 | 0.469 | 0.777 | 0.420 |
| | 720 | 0.469 | 0.285 | 0.467 | 0.302 | 0.499 | 0.321 | 0.645 | 0.394 | 0.640 | 0.350 | 0.626 | 0.382 | 0.632 | 0.396 | 0.658 | 0.407 | 0.660 | 0.408 | 0.896 | 0.473 | 0.864 | 0.472 |
| | Avg | 0.433 | 0.265 | 0.428 | 0.282 | 0.465 | 0.304 | 0.625 | 0.383 | 0.620 | 0.336 | 0.610 | 0.376 | 0.621 | 0.396 | 0.622 | 0.392 | 0.628 | 0.379 | 0.878 | 0.469 | 0.764 | 0.416 |
| | Me | 0.432 | 0.265 | 0.425 | 0.280 | 0.462 | 0.302 | 0.625 | 0.384 | 0.623 | 0.336 | 0.613 | 0.378 | 0.622 | 0.396 | 0.614 | 0.389 | 0.619 | 0.385 | 0.875 | 0.469 | 0.748 | 0.406 |
| Weather | 96 | 0.152 | 0.202 | 0.174 | 0.214 | 0.163 | 0.207 | 0.196 | 0.255 | 0.172 | 0.220 | 0.217 | 0.296 | 0.197 | 0.281 | 0.182 | 0.242 | 0.266 | 0.336 | 0.622 | 0.556 | 0.300 | 0.384 |
| | 192 | 0.200 | 0.250 | 0.221 | 0.254 | 0.214 | 0.253 | 0.237 | 0.296 | 0.219 | 0.261 | 0.276 | 0.336 | 0.237 | 0.312 | 0.227 | 0.287 | 0.307 | 0.367 | 0.739 | 0.624 | 0.598 | 0.544 |
| | 336 | 0.261 | 0.305 | 0.278 | 0.296 | 0.271 | 0.294 | 0.283 | 0.335 | 0.280 | 0.306 | 0.339 | 0.380 | 0.298 | 0.353 | 0.282 | 0.334 | 0.359 | 0.395 | 1.004 | 0.753 | 0.578 | 0.523 |
| | 720 | 0.325 | 0.349 | 0.358 | 0.349 | 0.360 | 0.346 | 0.345 | 0.381 | 0.365 | 0.359 | 0.403 | 0.428 | 0.352 | 0.390 | 0.352 | 0.386 | 0.419 | 0.428 | 1.420 | 0.934 | 1.059 | 0.741 |
| | Avg | 0.234 | 0.276 | 0.258 | 0.279 | 0.249 | 0.275 | 0.265 | 0.317 | 0.259 | 0.287 | 0.309 | 0.360 | 0.271 | 0.334 | 0.261 | 0.312 | 0.338 | 0.382 | 0.946 | 0.717 | 0.634 | 0.548 |
| | Me | 0.230 | 0.277 | 0.250 | 0.275 | 0.243 | 0.274 | 0.260 | 0.316 | 0.250 | 0.284 | 0.308 | 0.358 | 0.268 | 0.333 | 0.255 | 0.311 | 0.333 | 0.381 | 0.872 | 0.689 | 0.588 | 0.534 |
| ETTh1 | 96 | 0.386 | 0.400 | 0.386 | 0.405 | 0.392 | 0.409 | 0.386 | 0.400 | 0.384 | 0.402 | 0.376 | 0.419 | 0.494 | 0.479 | 0.424 | 0.432 | 0.449 | 0.459 | 0.664 | 0.612 | 0.865 | 0.713 |
| | 192 | 0.440 | 0.431 | 0.441 | 0.436 | 0.445 | 0.436 | 0.437 | 0.432 | 0.436 | 0.429 | 0.420 | 0.448 | 0.538 | 0.504 | 0.475 | 0.462 | 0.500 | 0.482 | 0.790 | 0.681 | 1.008 | 0.792 |
| | 336 | 0.478 | 0.451 | 0.487 | 0.458 | 0.482 | 0.456 | 0.481 | 0.459 | 0.491 | 0.469 | 0.459 | 0.465 | 0.574 | 0.521 | 0.518 | 0.488 | 0.521 | 0.496 | 0.891 | 0.738 | 1.107 | 0.809 |
| | 720 | 0.476 | 0.468 | 0.503 | 0.491 | 0.489 | 0.480 | 0.519 | 0.516 | 0.521 | 0.500 | 0.506 | 0.507 | 0.562 | 0.535 | 0.547 | 0.533 | 0.514 | 0.512 | 0.963 | 0.782 | 1.181 | 0.865 |
| | Avg | 0.445 | 0.437 | 0.454 | 0.447 | 0.452 | 0.445 | 0.456 | 0.452 | 0.458 | 0.450 | 0.440 | 0.460 | 0.542 | 0.510 | 0.491 | 0.479 | 0.496 | 0.487 | 0.827 | 0.703 | 1.040 | 0.795 |
| | Me | 0.458 | 0.441 | 0.464 | 0.447 | 0.464 | 0.446 | 0.459 | 0.446 | 0.464 | 0.449 | 0.440 | 0.457 | 0.550 | 0.513 | 0.497 | 0.475 | 0.507 | 0.489 | 0.841 | 0.710 | 1.058 | 0.801 |
| ETTh2 | 96 | 0.299 | 0.350 | 0.297 | 0.349 | 0.305 | 0.353 | 0.333 | 0.387 | 0.340 | 0.374 | 0.358 | 0.397 | 0.340 | 0.391 | 0.397 | 0.437 | 0.346 | 0.388 | 0.645 | 0.597 | 3.755 | 1.525 |
| | 192 | 0.364 | 0.391 | 0.380 | 0.400 | 0.382 | 0.401 | 0.477 | 0.476 | 0.402 | 0.414 | 0.429 | 0.439 | 0.430 | 0.439 | 0.520 | 0.504 | 0.456 | 0.452 | 0.788 | 0.683 | 5.602 | 1.931 |
| | 336 | 0.409 | 0.427 | 0.428 | 0.432 | 0.434 | 0.445 | 0.594 | 0.541 | 0.452 | 0.452 | 0.496 | 0.487 | 0.485 | 0.479 | 0.626 | 0.559 | 0.482 | 0.486 | 0.907 | 0.747 | 4.721 | 1.835 |
| | 720 | 0.421 | 0.443 | 0.427 | 0.445 | 0.424 | 0.444 | 0.831 | 0.657 | 0.462 | 0.468 | 0.463 | 0.474 | 0.500 | 0.497 | 0.863 | 0.672 | 0.515 | 0.511 | 0.963 | 0.783 | 3.647 | 1.625 |
| | Avg | 0.373 | 0.402 | 0.383 | 0.407 | 0.386 | 0.411 | 0.559 | 0.515 | 0.414 | 0.427 | 0.437 | 0.449 | 0.439 | 0.452 | 0.602 | 0.543 | 0.450 | 0.459 | 0.826 | 0.703 | 4.431 | 1.729 |
| | Me | 0.386 | 0.409 | 0.404 | 0.416 | 0.403 | 0.423 | 0.536 | 0.509 | 0.427 | 0.433 | 0.446 | 0.457 | 0.458 | 0.459 | 0.573 | 0.532 | 0.469 | 0.469 | 0.848 | 0.715 | 4.238 | 1.730 |
| ETTm1 | 96 | 0.324 | 0.356 | 0.334 | 0.368 | 0.336 | 0.369 | 0.345 | 0.372 | 0.338 | 0.375 | 0.379 | 0.419 | 0.375 | 0.398 | 0.374 | 0.409 | 0.505 | 0.475 | 0.543 | 0.510 | 0.672 | 0.571 |
| | 192 | 0.366 | 0.383 | 0.377 | 0.391 | 0.374 | 0.387 | 0.380 | 0.389 | 0.374 | 0.387 | 0.426 | 0.441 | 0.408 | 0.410 | 0.400 | 0.407 | 0.553 | 0.496 | 0.557 | 0.537 | 0.795 | 0.669 |
| | 336 | 0.395 | 0.403 | 0.426 | 0.420 | 0.408 | 0.407 | 0.413 | 0.413 | 0.410 | 0.411 | 0.445 | 0.459 | 0.435 | 0.428 | 0.438 | 0.438 | 0.621 | 0.537 | 0.754 | 0.655 | 1.212 | 0.871 |
| | 720 | 0.467 | 0.440 | 0.491 | 0.459 | 0.477 | 0.442 | 0.474 | 0.453 | 0.478 | 0.450 | 0.543 | 0.490 | 0.499 | 0.462 | 0.527 | 0.502 | 0.671 | 0.561 | 0.908 | 0.724 | 1.166 | 0.823 |
| | Avg | 0.388 | 0.395 | 0.407 | 0.410 | 0.399 | 0.401 | 0.403 | 0.407 | 0.400 | 0.406 | 0.448 | 0.452 | 0.429 | 0.425 | 0.435 | 0.437 | 0.588 | 0.517 | 0.691 | 0.607 | 0.961 | 0.734 |
| | Me | 0.380 | 0.383 | 0.402 | 0.406 | 0.391 | 0.397 | 0.397 | 0.401 | 0.392 | 0.399 | 0.436 | 0.450 | 0.422 | 0.419 | 0.419 | 0.423 | 0.587 | 0.517 | 0.656 | 0.596 | 0.981 | 0.746 |
| ETTm2 | 96 | 0.180 | 0.262 | 0.180 | 0.264 | 0.184 | 0.267 | 0.193 | 0.292 | 0.187 | 0.267 | 0.203 | 0.287 | 0.189 | 0.280 | 0.209 | 0.308 | 0.255 | 0.339 | 0.435 | 0.507 | 0.365 | 0.453 |
| | 192 | 0.246 | 0.306 | 0.250 | 0.309 | 0.252 | 0.307 | 0.284 | 0.362 | 0.249 | 0.309 | 0.269 | 0.328 | 0.253 | 0.319 | 0.311 | 0.382 | 0.281 | 0.340 | 0.730 | 0.673 | 0.533 | 0.563 |
| | 336 | 0.307 | 0.340 | 0.311 | 0.348 | 0.314 | 0.345 | 0.369 | 0.427 | 0.321 | 0.351 | 0.325 | 0.366 | 0.314 | 0.357 | 0.442 | 0.446 | 0.339 | 0.372 | 1.201 | 0.845 | 1.363 | 0.887 |
| | 720 | 0.408 | 0.403 | 0.412 | 0.407 | 0.412 | 0.402 | 0.554 | 0.522 | 0.408 | 0.403 | 0.421 | 0.415 | 0.414 | 0.413 | 0.675 | 0.587 | 0.433 | 0.432 | 3.625 | 1.451 | 3.379 | 1.338 |
| | Avg | 0.285 | 0.327 | 0.288 | 0.332 | 0.291 | 0.330 | 0.350 | 0.401 | 0.291 | 0.333 | 0.305 | 0.349 | 0.293 | 0.342 | 0.409 | 0.436 | 0.327 | 0.371 | 1.498 | 0.869 | 1.410 | 0.810 |
| | Me | 0.276 | 0.323 | 0.281 | 0.329 | 0.283 | 0.326 | 0.327 | 0.395 | 0.285 | 0.330 | 0.297 | 0.347 | 0.284 | 0.338 | 0.377 | 0.424 | 0.310 | 0.356 | 0.966 | 0.759 | 0.948 | 0.725 |
| Exchange | 96 | 0.084 | 0.207 | 0.086 | 0.206 | 0.086 | 0.206 | 0.088 | 0.218 | 0.107 | 0.234 | 0.148 | 0.278 | 0.085 | 0.204 | 0.116 | 0.262 | 0.197 | 0.323 | 1.748 | 1.105 | 0.847 | 0.752 |
| | 192 | 0.188 | 0.319 | 0.177 | 0.299 | 0.176 | 0.299 | 0.176 | 0.315 | 0.226 | 0.334 | 0.271 | 0.380 | 0.182 | 0.303 | 0.215 | 0.359 | 0.300 | 0.369 | 1.874 | 1.151 | 1.204 | 0.895 |
| | 336 | 0.329 | 0.425 | 0.331 | 0.417 | 0.330 | 0.416 | 0.313 | 0.427 | 0.367 | 0.448 | 0.460 | 0.500 | 0.348 | 0.428 | 0.377 | 0.466 | 0.509 | 0.524 | 1.943 | 1.172 | 1.672 | 1.036 |
| | 720 | 0.779 | 0.670 | 0.847 | 0.691 | 0.828 | 0.689 | 0.839 | 0.695 | 0.964 | 0.746 | 1.195 | 0.841 | 1.025 | 0.774 | 0.831 | 0.699 | 1.447 | 0.941 | 2.085 | 1.206 | 2.478 | 1.310 |
| | Avg | 0.345 | 0.405 | 0.360 | 0.403 | 0.355 | 0.403 | 0.354 | 0.414 | 0.416 | 0.443 | 0.519 | 0.500 | 0.410 | 0.427 | 0.385 | 0.447 | 0.613 | 0.539 | 1.913 | 1.159 | 1.550 | 0.998 |
| | Me | 0.258 | 0.372 | 0.254 | 0.358 | 0.253 | 0.358 | 0.245 | 0.371 | 0.297 | 0.396 | 0.366 | 0.440 | 0.265 | 0.366 | 0.296 | 0.413 | 0.405 | 0.447 | 1.909 | 1.162 | 1.438 | 0.966 |
| 1st Count | | 70 | | 12 | | 9 | | 4 | | 2 | | 5 | | 1 | | 0 | | 0 | | 0 | | 0 | |
| 2st Count | | 15 | | 28 | | 48 | | 4 | | 5 | | 0 | | 3 | | 0 | | 0 | | 0 | | 0 | |
| Avg 1st Count | | 13 | | 2 | | 1 | | 0 | | 0 | | 0 | | 0 | | 0 | | 0 | | 0 | | 0 | |
| Me 1st Count | | 12 | | 2 | | 2 | | 1 | | 0 | | 0 | | 0 | | 0 | | 0 | | 0 | | 0 | |

Table 4: The complete results for LTSF. The results of 4 different prediction lengths of different models are listed in the table. The look-back window sizes are set to 96 for all datasets. We also calculate the average (Avg) and median(Me) of the results for the 4 prediction lengths and the number of optimal values obtained by different models.

| | Models / Metric | TVDN MSE | TVDN MAE | FITS 2024 MSE | FITS 2024 MAE | WITRAN 2024 MSE | WITRAN 2024 MAE | DLinear 2023 MSE | DLinear 2023 MAE | TimesNet 2022 MSE | TimesNet 2022 MAE | FEDformer 2022b MSE | FEDformer 2022b MAE | ETSformer 2022 MSE | ETSformer 2022 MAE | LightTS 2022 MSE | LightTS 2022 MAE | Autoformer 2021 MSE | Autoformer 2021 MAE | Pyraformer 2021 MSE | Pyraformer 2021 MAE | Informer 2021 MSE | Informer 2021 MAE |
|---|---|---|---|---|---|---|---|---|---|---|---|---|---|---|---|---|---|---|---|---|---|---|---|
| Electricity | 96 | **0.132** | **0.226** | 0.293 | 0.401 | 0.237 | 0.335 | 0.197 | 0.282 | 0.168 | 0.272 | 0.193 | 0.308 | 0.187 | 0.304 | 0.207 | 0.307 | 0.201 | 0.317 | 0.386 | 0.449 | 0.274 | 0.368 |
| Electricity | 192 | **0.153** | **0.250** | 0.268 | 0.378 | 0.258 | 0.350 | 0.196 | 0.285 | 0.184 | 0.289 | 0.201 | 0.315 | 0.199 | 0.315 | 0.213 | 0.316 | 0.222 | 0.334 | 0.378 | 0.443 | 0.296 | 0.386 |
| Electricity | 336 | **0.164** | **0.264** | 0.355 | 0.452 | 0.273 | 0.362 | 0.209 | 0.301 | 0.198 | 0.300 | 0.214 | 0.329 | 0.212 | 0.329 | 0.230 | 0.333 | 0.231 | 0.338 | 0.376 | 0.443 | 0.300 | 0.394 |
| Electricity | 720 | **0.186** | **0.284** | 0.416 | 0.498 | 0.300 | 0.382 | 0.245 | 0.333 | 0.220 | 0.320 | 0.246 | 0.355 | 0.233 | 0.245 | 0.265 | 0.360 | 0.254 | 0.361 | 0.376 | 0.445 | 0.373 | 0.439 |
| Electricity | Avg | **0.158** | **0.256** | 0.333 | 0.432 | 0.267 | 0.357 | 0.212 | 0.300 | 0.192 | 0.295 | 0.214 | 0.327 | 0.208 | 0.323 | 0.229 | 0.329 | 0.227 | 0.338 | 0.379 | 0.445 | 0.311 | 0.397 |
| Electricity | Me | **0.158** | **0.257** | 0.324 | 0.427 | 0.265 | 0.356 | 0.203 | 0.293 | 0.191 | 0.295 | 0.208 | 0.322 | 0.206 | 0.322 | 0.222 | 0.325 | 0.227 | 0.336 | 0.377 | 0.444 | 0.298 | 0.390 |
| Traffic | 96 | **0.401** | **0.248** | 0.898 | 0.572 | 1.037 | 0.441 | 0.650 | 0.396 | 0.593 | 0.321 | 0.587 | 0.366 | 0.607 | 0.392 | 0.615 | 0.391 | 0.613 | 0.388 | 0.867 | 0.468 | 0.719 | 0.391 |
| Traffic | 192 | **0.427** | **0.259** | 0.763 | 0.522 | 1.061 | 0.455 | 0.598 | 0.370 | 0.617 | 0.336 | 0.604 | 0.373 | 0.621 | 0.399 | 0.601 | 0.382 | 0.616 | 0.382 | 0.869 | 0.467 | 0.696 | 0.379 |
| Traffic | 336 | **0.438** | **0.271** | 0.894 | 0.608 | 1.095 | 0.470 | 0.605 | 0.373 | 0.629 | 0.336 | 0.621 | 0.383 | 0.622 | 0.399 | 0.613 | 0.386 | 0.622 | 0.337 | 0.881 | 0.469 | 0.777 | 0.420 |
| Traffic | 720 | **0.469** | **0.285** | 1.019 | 0.646 | 1.121 | 0.474 | 0.645 | 0.394 | 0.640 | 0.350 | 0.626 | 0.382 | 0.632 | 0.396 | 0.658 | 0.407 | 0.660 | 0.408 | 0.896 | 0.473 | 0.864 | 0.472 |
| Traffic | Avg | **0.433** | **0.265** | 0.894 | 0.587 | 1.079 | 0.460 | 0.625 | 0.383 | 0.620 | 0.336 | 0.610 | 0.376 | 0.621 | 0.396 | 0.622 | 0.392 | 0.628 | 0.379 | 0.878 | 0.469 | 0.764 | 0.416 |
| Traffic | Me | **0.432** | **0.265** | 0.879 | 0.597 | 1.078 | 0.463 | 0.625 | 0.384 | 0.623 | 0.336 | 0.613 | 0.378 | 0.622 | 0.396 | 0.614 | 0.389 | 0.619 | 0.385 | 0.875 | 0.469 | 0.748 | 0.406 |
| Weather | 96 | **0.152** | **0.202** | 0.174 | 0.214 | 0.178 | 0.223 | 0.196 | 0.255 | 0.172 | 0.220 | 0.217 | 0.296 | 0.197 | 0.281 | 0.182 | 0.242 | 0.266 | 0.336 | 0.622 | 0.556 | 0.300 | 0.384 |
| Weather | 192 | **0.200** | **0.250** | 0.221 | 0.254 | 0.223 | 0.261 | 0.237 | 0.296 | 0.219 | 0.261 | 0.276 | 0.336 | 0.237 | 0.312 | 0.227 | 0.287 | 0.307 | 0.367 | 0.739 | 0.624 | 0.598 | 0.544 |
| Weather | 336 | **0.261** | **0.305** | 0.278 | 0.309 | 0.288 | 0.309 | 0.283 | 0.335 | 0.280 | 0.306 | 0.339 | 0.380 | 0.298 | 0.353 | 0.282 | 0.334 | 0.359 | 0.395 | 1.004 | 0.753 | 0.578 | 0.523 |
| Weather | 720 | **0.325** | **0.349** | 0.358 | 0.349 | 0.372 | 0.363 | 0.345 | 0.381 | 0.365 | 0.359 | 0.403 | 0.428 | 0.352 | 0.390 | 0.352 | 0.386 | 0.419 | 0.428 | 1.420 | 0.934 | 1.059 | 0.741 |
| Weather | Avg | **0.234** | **0.276** | 0.258 | 0.278 | 0.265 | 0.289 | 0.265 | 0.317 | 0.259 | 0.287 | 0.309 | 0.360 | 0.271 | 0.334 | 0.261 | 0.312 | 0.338 | 0.382 | 0.946 | 0.717 | 0.634 | 0.548 |
| Weather | Me | **0.230** | **0.277** | 0.250 | 0.275 | 0.255 | 0.285 | 0.260 | 0.316 | 0.250 | 0.284 | 0.308 | 0.358 | 0.268 | 0.333 | 0.255 | 0.311 | 0.333 | 0.381 | 0.872 | 0.689 | 0.588 | 0.534 |
| ETTh1 | 96 | 0.386 | **0.400** | 0.381 | 0.391 | 0.414 | 0.419 | 0.386 | **0.400** | 0.384 | 0.402 | **0.376** | 0.419 | 0.494 | 0.479 | 0.424 | 0.432 | 0.449 | 0.459 | 0.664 | 0.612 | 0.865 | 0.713 |
| ETTh1 | 192 | 0.440 | 0.431 | 0.443 | 0.422 | 0.464 | 0.448 | 0.437 | 0.432 | 0.436 | **0.429** | **0.420** | 0.439 | 0.538 | 0.504 | 0.475 | 0.462 | 0.500 | 0.482 | 0.790 | 0.681 | 1.008 | 0.792 |
| ETTh1 | 336 | 0.478 | **0.451** | 0.474 | 0.446 | 0.516 | 0.478 | 0.481 | 0.459 | 0.477 | 0.456 | **0.459** | 0.465 | 0.574 | 0.521 | 0.518 | 0.488 | 0.521 | 0.496 | 0.891 | 0.738 | 1.107 | 0.809 |
| ETTh1 | 720 | 0.476 | **0.468** | 0.464 | 0.463 | 0.538 | 0.509 | 0.519 | 0.516 | 0.521 | 0.500 | **0.459** | 0.474 | 0.562 | 0.535 | 0.547 | 0.533 | 0.514 | 0.512 | 0.963 | 0.782 | 1.181 | 0.865 |
| ETTh1 | Avg | 0.445 | **0.437** | 0.438 | 0.431 | 0.483 | 0.464 | 0.456 | 0.452 | 0.444 | 0.447 | **0.429** | 0.449 | 0.542 | 0.510 | 0.491 | 0.479 | 0.496 | 0.487 | 0.827 | 0.703 | 1.040 | 0.795 |
| ETTh1 | Me | 0.458 | **0.441** | 0.459 | 0.434 | 0.490 | 0.463 | 0.459 | 0.446 | 0.456 | 0.445 | **0.440** | 0.452 | 0.550 | 0.513 | 0.497 | 0.475 | 0.507 | 0.489 | 0.841 | 0.710 | 1.058 | 0.801 |
| ETTh2 | 96 | 0.299 | 0.350 | **0.290** | **0.339** | 0.325 | 0.364 | 0.333 | 0.387 | 0.340 | 0.374 | 0.358 | 0.397 | 0.340 | 0.391 | 0.397 | 0.437 | 0.346 | 0.388 | 0.645 | 0.597 | 3.755 | 1.525 |
| ETTh2 | 192 | **0.364** | **0.391** | 0.375 | 0.388 | 0.433 | 0.427 | 0.477 | 0.476 | 0.402 | 0.414 | 0.429 | 0.439 | 0.430 | 0.439 | 0.520 | 0.504 | 0.456 | 0.452 | 0.788 | 0.683 | 5.602 | 1.931 |
| ETTh2 | 336 | **0.409** | **0.427** | 0.414 | 0.425 | 0.471 | 0.457 | 0.594 | 0.541 | 0.452 | 0.452 | 0.496 | 0.487 | 0.485 | 0.479 | 0.626 | 0.559 | 0.482 | 0.486 | 0.907 | 0.747 | 4.721 | 1.835 |
| ETTh2 | 720 | **0.421** | **0.443** | 0.419 | 0.437 | 0.499 | 0.480 | 0.831 | 0.657 | 0.424 | 0.444 | 0.463 | 0.474 | 0.500 | 0.497 | 0.863 | 0.672 | 0.515 | 0.511 | 0.963 | 0.783 | 3.647 | 1.625 |
| ETTh2 | Avg | **0.373** | **0.402** | 0.375 | 0.397 | 0.432 | 0.432 | 0.559 | 0.515 | 0.414 | 0.427 | 0.437 | 0.449 | 0.439 | 0.452 | 0.602 | 0.543 | 0.450 | 0.459 | 0.826 | 0.703 | 4.431 | 1.729 |
| ETTh2 | Me | **0.386** | **0.409** | 0.395 | 0.406 | 0.452 | 0.442 | 0.536 | 0.509 | 0.427 | 0.433 | 0.446 | 0.457 | 0.458 | 0.459 | 0.573 | 0.532 | 0.469 | 0.469 | 0.848 | 0.715 | 4.238 | 1.730 |
| ETTm1 | 96 | **0.324** | **0.356** | 0.351 | 0.370 | 0.375 | 0.402 | 0.345 | 0.372 | 0.338 | 0.375 | 0.379 | 0.419 | 0.375 | 0.398 | 0.374 | 0.409 | 0.505 | 0.475 | 0.543 | 0.510 | 0.672 | 0.571 |
| ETTm1 | 192 | **0.366** | **0.383** | 0.392 | 0.393 | 0.427 | 0.434 | 0.380 | 0.389 | 0.374 | 0.387 | 0.426 | 0.441 | 0.408 | 0.410 | 0.400 | 0.407 | 0.553 | 0.496 | 0.557 | 0.537 | 0.795 | 0.669 |
| ETTm1 | 336 | **0.395** | **0.403** | 0.424 | 0.413 | 0.455 | 0.452 | 0.413 | 0.413 | 0.408 | 0.407 | 0.445 | 0.459 | 0.435 | 0.428 | 0.438 | 0.438 | 0.621 | 0.537 | 0.754 | 0.655 | 1.212 | 0.871 |
| ETTm1 | 720 | **0.467** | **0.440** | 0.485 | 0.448 | 0.527 | 0.488 | 0.474 | 0.453 | 0.478 | 0.442 | 0.543 | 0.490 | 0.499 | 0.462 | 0.527 | 0.502 | 0.671 | 0.561 | 0.908 | 0.724 | 1.166 | 0.823 |
| ETTm1 | Avg | **0.388** | **0.395** | 0.413 | 0.406 | 0.446 | 0.444 | 0.403 | 0.407 | 0.400 | 0.403 | 0.448 | 0.452 | 0.429 | 0.425 | 0.435 | 0.439 | 0.588 | 0.517 | 0.691 | 0.607 | 0.961 | 0.734 |
| ETTm1 | Me | **0.380** | **0.393** | 0.408 | 0.403 | 0.441 | 0.443 | 0.397 | 0.401 | 0.391 | 0.397 | 0.436 | 0.450 | 0.422 | 0.419 | 0.419 | 0.423 | 0.587 | 0.517 | 0.656 | 0.596 | 0.981 | 0.746 |
| ETTm2 | 96 | 0.180 | 0.262 | **0.181** | 0.264 | 0.191 | 0.272 | 0.193 | 0.292 | **0.187** | 0.267 | 0.203 | 0.287 | 0.189 | 0.280 | 0.209 | 0.308 | 0.255 | 0.339 | 0.435 | 0.507 | 0.365 | 0.453 |
| ETTm2 | 192 | **0.246** | 0.306 | 0.246 | 0.304 | 0.261 | 0.316 | 0.284 | 0.362 | 0.249 | 0.307 | 0.269 | 0.328 | 0.253 | 0.319 | 0.311 | 0.382 | 0.281 | 0.340 | 0.730 | 0.673 | 0.533 | 0.563 |
| ETTm2 | 336 | **0.307** | 0.340 | 0.306 | 0.341 | 0.330 | 0.358 | 0.369 | 0.427 | 0.321 | 0.351 | 0.325 | 0.366 | 0.314 | 0.357 | 0.442 | 0.446 | 0.339 | 0.372 | 1.201 | 0.845 | 1.363 | 0.887 |
| ETTm2 | 720 | **0.408** | 0.403 | 0.407 | 0.397 | 0.450 | 0.427 | 0.554 | 0.522 | 0.408 | 0.403 | 0.421 | 0.415 | 0.414 | 0.413 | 0.675 | 0.587 | 0.433 | 0.432 | 3.625 | 1.451 | 3.379 | 1.338 |
| ETTm2 | Avg | **0.285** | 0.327 | 0.285 | 0.327 | 0.308 | 0.343 | 0.350 | 0.401 | 0.291 | 0.333 | 0.305 | 0.349 | 0.293 | 0.342 | 0.409 | 0.436 | 0.327 | 0.371 | 1.498 | 0.869 | 1.410 | 0.810 |
| ETTm2 | Me | **0.276** | 0.323 | 0.276 | 0.323 | 0.296 | 0.337 | 0.327 | 0.395 | 0.285 | 0.330 | 0.297 | 0.347 | 0.284 | 0.338 | 0.377 | 0.424 | 0.310 | 0.356 | 0.966 | 0.759 | 0.948 | 0.725 |
| | 1st Count | **73** | | 3 | | 0 | | 1 | | 1 | | 6 | | 0 | | 0 | | 0 | | 0 | | 0 | |
| | 2st Count | 4 | | **33** | | 0 | | 11 | | 33 | | 1 | | 0 | | 0 | | 0 | | 0 | | 0 | |
| | Avg 1st Count | **12** | | 0 | | 0 | | 0 | | 0 | | 1 | | 0 | | 0 | | 0 | | 0 | | 0 | |
| | Me 1st Count | **12** | | 0 | | 0 | | 0 | | 0 | | 1 | | 0 | | 0 | | 0 | | 0 | | 0 | |

## C  VISUALIZATION OF MAIN RESULTS

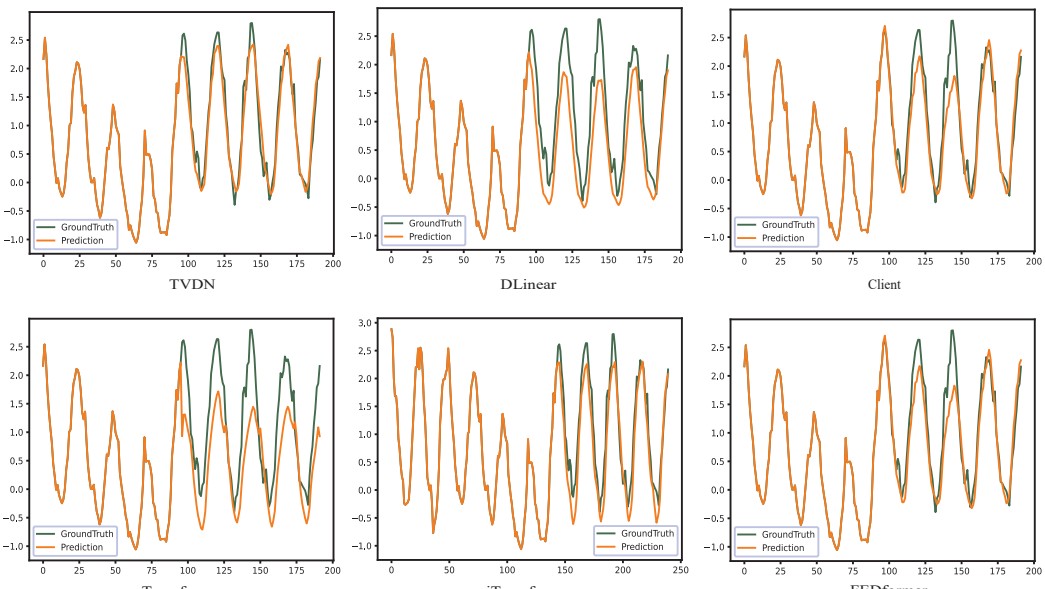

Figure 9: Visualization of the prediction results on the Electricity dataset, where TVDN predicts more accurately compared to other models in terms of better fitting the actual series.

# D    Performance with increasing lookback length

To investigate the impact of increasing lookback length on model performance, we conducted comparative experiments across different input sequence lengths (L). As shown in Figure 10, we evaluate TVDN against state-of-the-art baselines on the electricity dataset under both short-term (T=96) and long-term (T=720) forecasting scenarios.

Previous studies have observed that increasing lookback length does not necessarily improve forecasting performance in Transformer-based models, primarily due to distracted attention on growing input sequences (Zeng et al., 2023; Liu et al., 2024; Gao et al., 2023b). Our experimental results reveal distinct patterns: while traditional Transformer-based models show inconsistent performance with increased lookback lengths, TVDN demonstrates robust and improving performance as L increases from 24 to 720.

For T=96, TVDN's MSE steadily decreases, effectively utilizing longer historical information. PatchTST and DLinear also show improvements with increasing lookback lengths, but their performances are worse than TVDN. In contrast, Transformer, FEDformer, and Autoformer exhibit relatively unstable performance patterns, confirming the attention distraction phenomenon noted in previous works(Liu et al., 2024).

The advantage of TVDN becomes more pronounced in the long-term forecasting scenario (T=720). TVDN's performance is consistently better than the other models, including PatchTST and DLinear. While Autoformer and Transformer show significant fluctuations, particularly in the L=48 to L=96 range, TVDN maintains stable performance and achieves optimal results in the L=192-336 range. This demonstrates TVDN's superior capability in handling longer sequences decrease suffering from the attention distraction issues that plague traditional Transformer architectures.

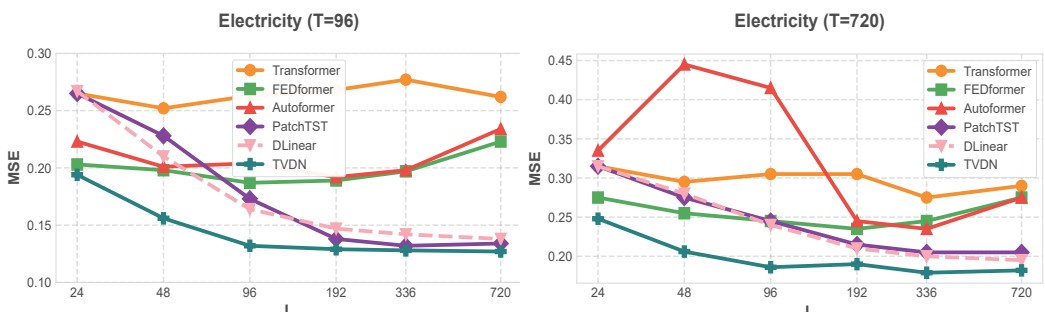

Figure 10: Performance comparison of TVDN against baseline models on the electricity dataset. Results are shown for two prediction lengths: T=96 (left) and T=720 (right). The x-axis represents different input sequence lengths (L), and the y-axis shows the Mean Square Error (MSE). TVDN consistently achieves lower MSE across different sequence lengths, particularly demonstrating better performance in long-term forecasting scenarios.

[tVnS]

# E  Robustness Analysis of TVDN Model

In this appendix, we present a comprehensive analysis of TVDN's robustness against different types of data perturbations commonly encountered in real-world applications. Specifically, we evaluate the model's performance under two major categories of data corruption: Gaussian noise and missing values. To assess TVDN's resilience to random disturbances, we conducted experiments by introducing Gaussian noise at various intensity levels (from 0.0 to 1.0). The noise was added to the input sequences following $x'_t = x_t + \epsilon$, where $\epsilon \sim \mathcal{N}(0, \sigma^2)$, $\sigma^2$ represents the noise level, $x_t$ is the original value at time $t$, and $x'_t$ is the corrupted value.

Table 5: Performance comparison of TVDN under different Gaussian noise levels (0.0-1.0), where noise level $\sigma$ represents the standard deviation of the additive Gaussian noise $x'_t = x_t + \epsilon$, $\epsilon \sim \mathcal{N}(0, \sigma^2)$. The evaluation metrics include MSE and MAE across multiple prediction horizons (96, 192, 336, and 720 steps), demonstrating the model's robustness against input perturbations.

| Models | | 96 steps | | 192 steps | | 336 steps | | 720 steps | |
| --- | --- | --- | --- | --- | --- | --- | --- | --- | --- |
| Metric | | MSE | MAE | MSE | MAE | MSE | MAE | MSE | MAE |
| | 0.0 | **0.132** | **0.226** | **0.153** | **0.250** | **0.164** | **0.264** | **0.186** | **0.284** |
| | 0.1 | 0.136 | 0.232 | 0.153 | 0.253 | 0.167 | 0.267 | 0.195 | 0.290 |
| | 0.2 | 0.136 | 0.237 | 0.156 | 0.258 | 0.171 | 0.276 | 0.186 | 0.286 |
| | 0.3 | 0.140 | 0.244 | 0.157 | 0.262 | 0.168 | 0.276 | 0.196 | 0.298 |
| Noise Level | 0.4 | 0.144 | 0.250 | 0.160 | 0.268 | 0.173 | 0.283 | 0.192 | 0.295 |
| | 0.5 | 0.147 | 0.255 | 0.164 | 0.273 | 0.173 | 0.282 | 0.193 | 0.300 |
| | 0.7 | 0.155 | 0.266 | 0.171 | 0.283 | 0.179 | 0.291 | 0.197 | 0.306 |
| | 1.0 | 0.165 | 0.281 | 0.184 | 0.297 | 0.191 | 0.306 | 0.209 | 0.319 |
| Average | | 0.144 | 0.249 | 0.162 | 0.268 | 0.173 | 0.281 | 0.194 | 0.297 |

The experimental results in Table 5 and Figure 11 demonstrate that the performance degradation follows a gradual trend as noise intensity increases. At low noise levels (0.1-0.3), the model maintains performance close to the baseline, with degradation limited to within 10%. Even at high noise levels (0.7-1.0), the increase in MSE and MAE remains within 25% of the baseline performance.

Table 6: Performance evaluation of TVDN under varying missing value rates (0.0-0.7), where missing rate represents the proportion of randomly masked values in the input sequence $x_t$. Results are measured using MSE and MAE across different prediction lengths (96, 192, 336, and 720 steps), illustrating the model's capability in handling incomplete time series data.

| Models | | 96 steps | | 192 steps | | 336 steps | | 720 steps | |
| --- | --- | --- | --- | --- | --- | --- | --- | --- | --- |
| Metric | | MSE | MAE | MSE | MAE | MSE | MAE | MSE | MAE |
| | 0.0 | **0.132** | **0.226** | **0.153** | **0.250** | **0.164** | **0.264** | **0.186** | **0.284** |
| | 0.1 | 0.140 | 0.241 | 0.152 | 0.256 | 0.165 | 0.271 | 0.185 | 0.288 |
| Missing Rate | 0.3 | 0.147 | 0.252 | 0.159 | 0.264 | 0.172 | 0.280 | 0.191 | 0.301 |
| | 0.5 | 0.156 | 0.263 | 0.169 | 0.276 | 0.179 | 0.289 | 0.198 | 0.307 |
| | 0.7 | 0.168 | 0.275 | 0.182 | 0.287 | 0.194 | 0.301 | 0.216 | 0.321 |
| Average | | 0.149 | 0.251 | 0.163 | 0.267 | 0.175 | 0.281 | 0.195 | 0.300 |

To evaluate TVDN's capability in handling incomplete data, we conducted experiments with missing values by randomly masking out portions of the input sequence at different rates (0.1 to 0.7). The results in Table 6 and Figure 12 show that the model demonstrates strong resilience to missing values. At moderate missing rates (0.1-0.3), the performance degradation is limited to within 10%, and even with 70% missing values, the model maintains reasonable prediction accuracy with performance degradation within 30% of the baseline.

The experimental results demonstrate TVDN's robust performance under both Gaussian noise and missing values. Several factors contribute to this resilience. First, the temporal-value decomposition mechanism helps isolate noise effects from the underlying temporal

patterns. Second, the multi-scale feature extraction enables the model to capture temporal dependencies at different granularities, reducing the impact of local perturbations. Third, the adaptive attention mechanism can effectively focus on more reliable segments of the input sequence. These findings suggest that TVDN is well-suited for real-world applications where data quality cannot be guaranteed.

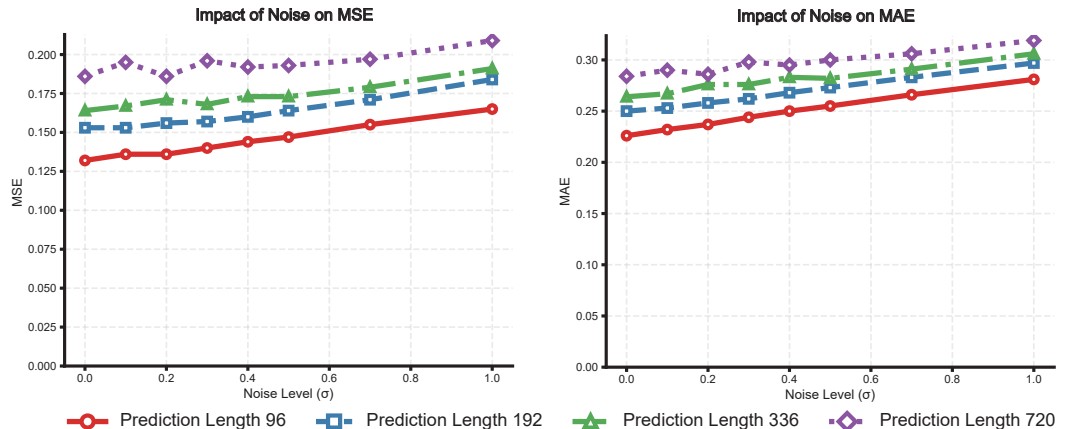

Figure 11: Impact of Gaussian noise on TVDN's prediction performance, where noise level $\sigma$ represents the standard deviation of the additive Gaussian noise $x'_t = x_t + \epsilon$, $\epsilon \sim \mathcal{N}(0, \sigma^2)$. A higher $\sigma$ indicates stronger noise perturbation on the original input sequence $x_t$. The left panel shows MSE and right panel shows MAE versus noise level (0.0 to 1.0) for prediction lengths of 96, 192, 336, and 720 time steps. Key findings: (1) MSE and MAE increase gradually with noise level; (2) Longer prediction horizons show higher error rates; (3) Performance degradation remains stable across noise levels; and (4) Performance gaps between prediction lengths remain consistent, demonstrating TVDN's robust handling of noisy data.

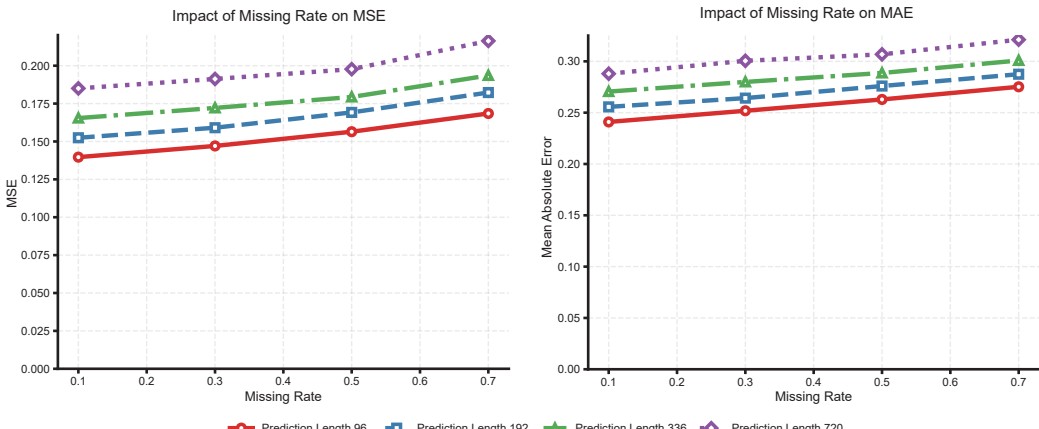

Figure 12: Impact of missing values on TVDN's prediction performance, where missing rate represents the proportion of randomly masked values in the input sequence. The left panel shows MSE and right panel shows MAE versus missing rate (0.1 to 0.7) for prediction lengths of 96, 192, 336, and 720 time steps. Key findings: (1) MSE and MAE show moderate increases with higher missing rates; (2) Performance remains stable even at 0.7 missing rate; (3) Shorter prediction horizons maintain better performance; and (4) Performance gaps between prediction lengths remain stable across missing rates, demonstrating TVDN's robust handling of incomplete data.

[Jesy]

# F   Transformer Limitations Analysis

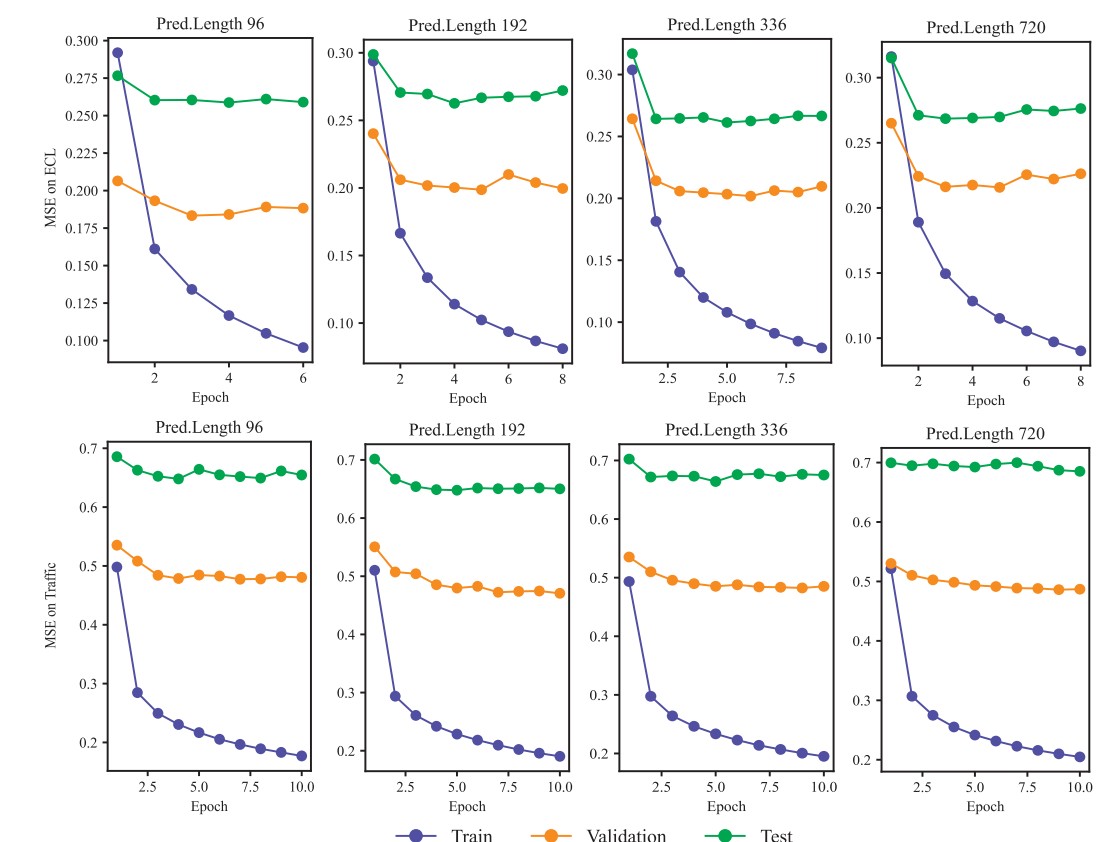

Figure 13: Observation of the model's loss trend on the Electricity and Traffic datasets. Training was fixed for 10 epochs with an early stopping tolerance of 3. Training was terminated upon exceeding this tolerance level.

In the context of time series prediction problems based on Transformer models, we can perceive the data-driven learning of the Transformer model as two distinct parts. The first part involves the encoder extracting valuable information from historical sequences through self-attention and feed-forward networks (FFNs). The second part is the decoder, which, in conjunction with the encoder's output, models the associative relationships of the target sequence.

To investigate which part primarily contributes to the Transformer model's benefits, we conducted an extreme experiment. This study tested the original Transformer model and a model using only the Transformer decoder on the Electricity and Traffic datasets. For the decoder-only model, we retained few historical sequences as start tokens for the Transformer's decoder, thereby minimizing the use of historical sequence information as much as possible.

As show in Figure13 When applying the original Transformer model to time series prediction, we observed significant overfitting. As shown in the figure, despite setting a relatively small learning rate $(1 \times 10^{-4})$, it's apparent that there's an early occurrence of the training set loss decreasing while the validation set loss increases. Moreover, the losses for both the validation and test sets stabilize quickly.

While the Transformer with historical information performs marginally better in most cases, the performance difference compared to the Transformer Decoder (without historical information) is insignificant. In certain cases, the Transformer Decoder even surpasses the full Transformer. This partially supports the hypothesis that the Transformer model may not effectively utilize historical information. This observation is consistent with previous findings

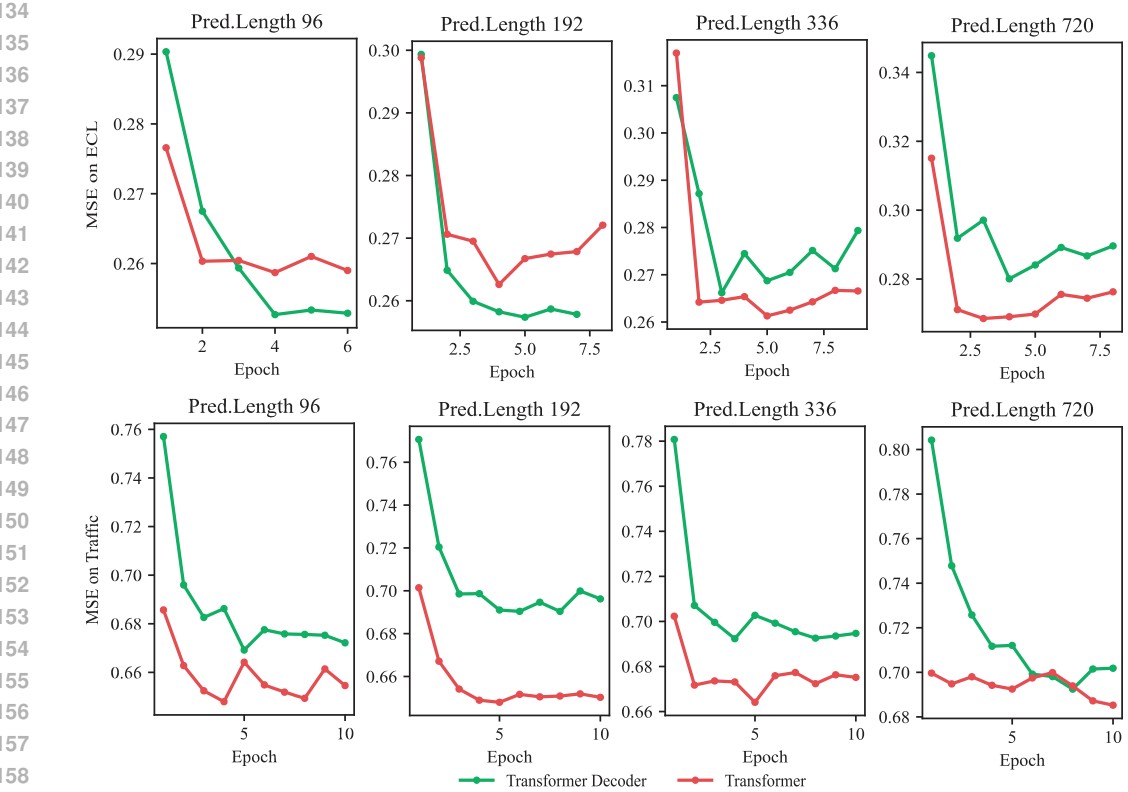

Figure 14: Comparative analysis of the original Transformer versus a decoder-only Transformer model on Electricity and Traffic datasets.

indicating that some Transformer-based models do not necessarily achieve better performance with an increased historical sequence length(Zeng et al., 2023; Liu et al., 2024; Gao et al., 2023b)

[tVnS]

As we can see, even when the Transformer model reduces the information from the historical sequence, its performance does not significantly decline. This suggests that modeling the temporal relationships in the prediction sequence is also crucial, which may be one of the reasons why the Transformer's performance remains stable.

## G   VISUALIZATION OF TVDN MODEL WEIGHT

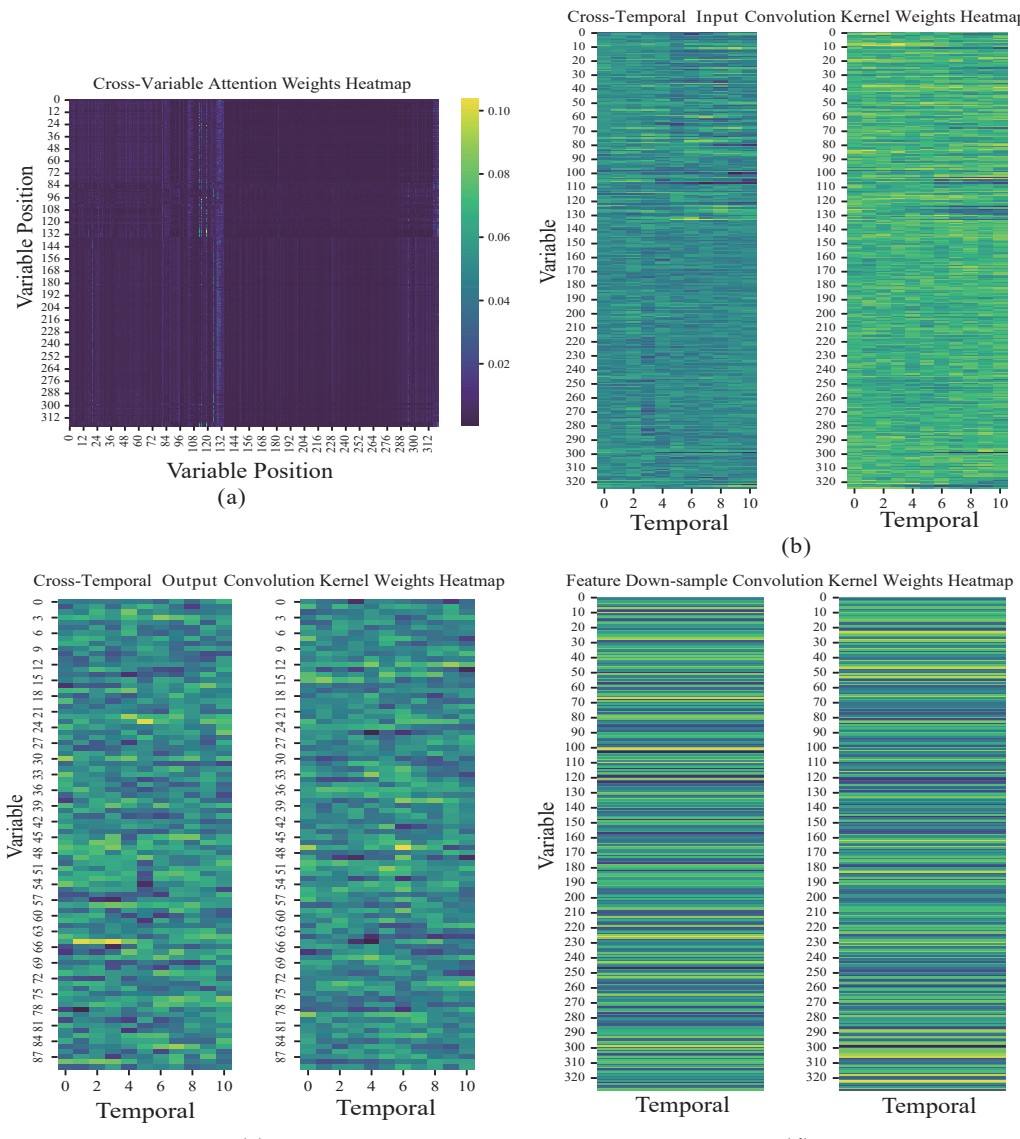

Figure 15: Visualization of TVDN Model Weights. (a) Heatmap of the attention matrix in CVE. (b) Heatmap of convolutional kernel weights in the local window of the input layer in CTE. (c) Convolutional kernel weights in the local window of the output layer in CTE. (d) Convolutional kernel weights for feature down-sampling in FDS.

## H  MOTIVATION FROM CROSS-VARIABLE LEARNING TO CROSS-TEMPORAL LEARNING

**Theoretical Motivation** Previous studies have highlighted that Cross-temporal Transformers are prone to bad local minima and are harder to converge to their true solutions Ilbert et al. (2024). Modeling cross-temporal relationships first can provide an unstable optimization starting point for subsequent cross-variable learning. In contrast, starting with cross-variable modeling helps establish a stable inter-variable relationship structure Liu et al. (2024); Gao et al. (2023b), which in turn provides a better optimization starting point for cross-temporal learning. This order increases the likelihood of convergence to the true solution and improves the overall performance of the model.

**Experimental Evidence** To validate the importance of this modeling order, we conducted experiments where the order of learning was reversed. The results clearly demonstrate that the proposed sequence of learning cross-variable relationships first (CVE) followed by cross-temporal relationships (CTE) outperforms the reversed order. The results are summarized in the Table 7.

[hsgf]

Table 7: Performance comparison of different learning orders on the ECL dataset. Results highlighted in red indicate the best performance for each prediction length.

| Prediction Length | CVE → CTE (Proposed) | | CTE → CVE (Reversed) | |
|---|---|---|---|---|
| | MSE | MAE | MSE | MAE |
| 96 | 0.132 | 0.226 | 0.191 | 0.295 |
| 192 | 0.153 | 0.250 | 0.194 | 0.293 |
| 336 | 0.164 | 0.264 | 0.194 | 0.294 |
| 720 | 0.186 | 0.284 | 0.228 | 0.321 |

## I  INSTANTANEOUS AND LAGGED EFFECTS DISCUSSION IN TVDN

In multivariate time series analysis, the temporal relationships between variables manifest as instantaneous and lagged effects. For example, in a biomedical time series, multiple physiological signals (e.g., heart rate and blood pressure) may be transiently correlated simultaneously. In some cases, there may be delayed effects between some variables. For example, the impact of temperature change on plant growth is usually gradual.

While our paper focuses on developing a general foundation model for various temporal data types, emphasizing the interaction between cross-variable and temporal dependencies, we should have explicitly discussed these temporal relationship types.

**Cross-variable learning: Can capture interactions between variables at same or different timesteps but overlook the specific time ordering.** In the cross-variable learning stage, the model can capture interactions between variables at different timesteps $(V_t^i$ and $V_{(t+\Delta)}^j)$, where $V_t^i$ represents the i-th variable at time t, and $V_{(t+\Delta)}^j$ represents the j-th variable at time $(t + \Delta)$. The temporal offset $\Delta$ allows the model to capture instantaneous effects (when $\Delta = 0$) and lagged effects (when $\Delta \neq 0$). This formulation maintains temporal invariance, meaning the model can identify relationships regardless of the specific time ordering of the variables.

**Cross-temporal learning: Incremental learning instead of siloed learning.** Our temporal learning component incrementally builds upon the cross-variable relationships identified in the first stage. Instead of treating these interactions in isolation, we integrate them to capture instantaneous and lagged effects better. This comprehensive approach ensures that our model effectively captures complex temporal dynamics, including direct and delayed influences between variables.

[hsgf]

# J    COMPARISON OF FOCUSING ON CROSS-VARIABLE LEARNING APPROACHES AND SHIFTING

Continuing to learn dependencies among variables results in a minimal decrease in MSE and can even lead to an increase in MSE, making overfitting more likely.

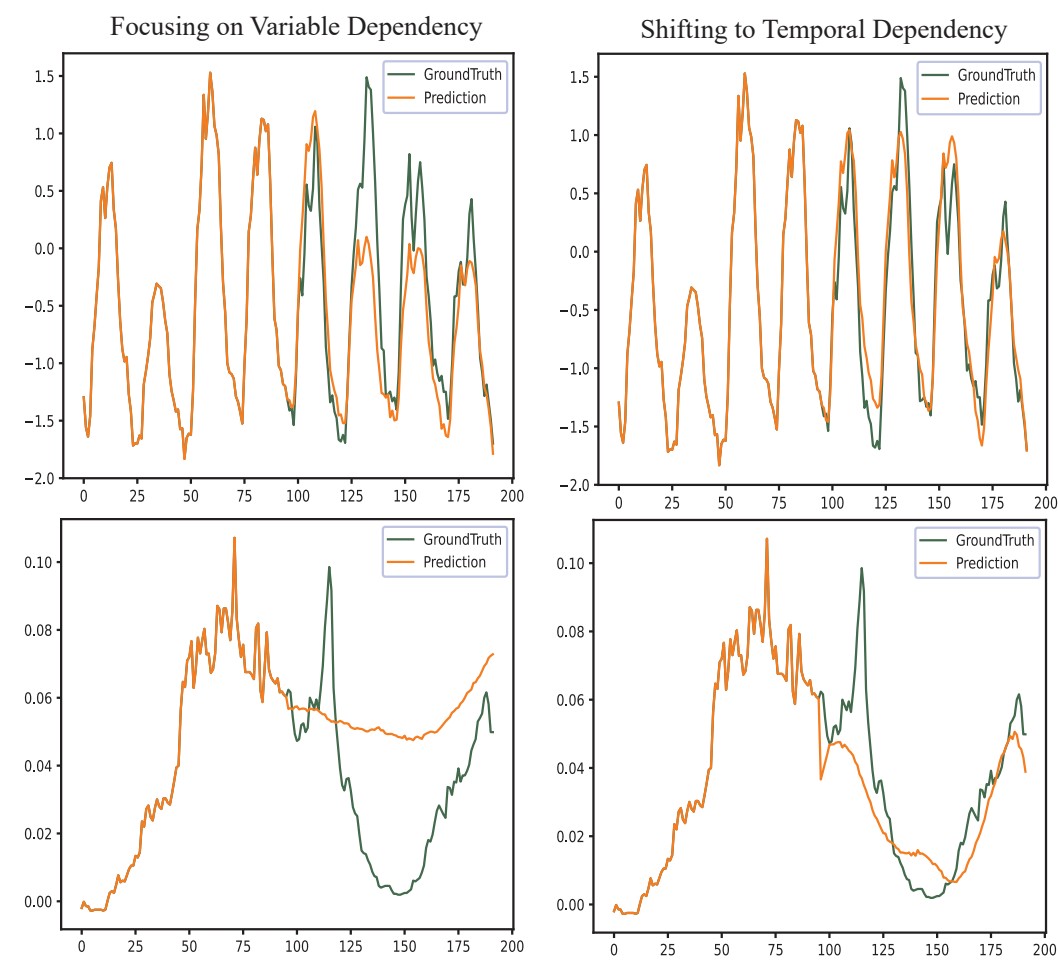

Figure 16: Comparison of focusing on Cross-variable learning approaches and shifting from Cross-variable learning to temporal dependency learning approaches. Visualization of prediction results on the ECL and Weather datasets. The latter shows a better fit for amplitude and trends.

# K   MODEL EFFICIENCY

[hsgf]

Table 8: Model efficiency comparison with state-of-the-art methods. FLOPs and parameters are measured on the ETTh1 dataset with prediction length 96. Time represents the average inference time per sample, and Memory denotes the peak memory usage during inference. The MSE values are averaged over all prediction lengths on ETTh1. Our TVDN achieves competitive performance (0.186 MSE) with moderate computation and memory costs (0.46G FLOPs, 50.25MB memory).

| Model | FLOPs (G) | Param (M) | Time (s) | Memory (MB) | MSE |
|---|---|---|---|---|---|
| TVDN (ours) | 0.46 | 1.44 | 0.0020 | 50.25 | 0.186 |
| iTransformer | 1.67 | 5.15 | 0.0019 | 62.06 | 0.225 |
| Client | 0.32 | 1.01 | 0.0016 | 46.33 | 0.209 |
| DLinear | 0.04 | 0.14 | 0.0003 | 42.94 | 0.245 |
| TimesNet | 612.79 | 150.37 | 0.0625 | 724.97 | 0.220 |
| FEDformer | 4.41 | 12.14 | 0.0298 | 246.33 | 0.246 |
| ETSformer | 0.85 | 6.57 | 0.0055 | 80.64 | 0.233 |
| LightTS | 0.10 | 0.33 | 0.0009 | 43.65 | 0.265 |
| Autoformer | 4.41 | 12.14 | 0.0107 | 221.52 | 0.254 |
| Pyraformer | 1.21 | 362.29 | 0.0039 | 1434.35 | 0.376 |
| Informer | 3.94 | 12.45 | 0.0055 | 218.42 | 0.373 |
| PatchTST | 25.73 | 10.74 | 0.0036 | 257.58 | 0.246 |

