# OpenReview forum: "Decoupling Variable and Temporal Dependencies: A Novel Approach for Multivariate Time Series Forecasting"
_ICLR.cc/2025/Conference — Submitted to ICLR 2025_

### Official Review · Reviewer_Jesy · 2024-11-03

**Soundness:** 3
**Presentation:** 3
**Contribution:** 3
**Rating:** 6
**Confidence:** 4

**Summary:**

This paper introduces a dual-phase deep learning network architecture called the Temporal-Variable Decoupling Network(TVDN), which decouples the modeling of variable dependencies from temporal dependencies. First, during the variable dependence learning phase, the Cross-Variable Encoder (CVE), a permutation-invariant model, completely disregards the temporal dependence of the sequence and only extracts cross-features between variables, generating an initial prediction sequence. Once the CVE stabilizes, the second phase shifts to temporal dependency learning. The Cross-Temporal Encoder (CTE) divides time series dependence into two parts: Historical Sequence Temporal Dependency (HSTD), and Prediction Sequence Temporal Dependency (PSTD). The outputs of HSTD and PSTD are then fused to correct the initial prediction from the variable dependence learning phase. The approach reduces the risk that overemphasizing temporal dependencies can overfit the model, and enables broader parameter space exploration. The experiment in this paper shows that TVDN achieves state-of-the-art (SOTA) performance in electricity, traffic, weather, four ETT, and exchange fields by comparing it with some of the latest Time-Series Forecasting Transformer (TSFT) methods.

**Strengths:**

1.	TVDN decouples the modeling of variable dependencies from temporal dependencies to reduce the interference between the two, enhance the utilization of temporal features, and improve forecasting accuracy.
2.	TVDN adopts a dual-phase architecture, starting with CVE to learn inter-variable dependencies and then shifting to CTE focused on temporal dependency learning. The two-phase approach effectively addresses the interference of variable and temporal learning, which avoids leading to a degradation of models’ performance.
3.	The proposed method demonstrates SOTA performance in real-world forecasting benchmarks by capturing complex dependencies across variables and time. This is achieved with minimal computational overhead, making the model efficient and scalable.

**Weaknesses:**

1.	The experiment focuses on predicting accuracy and sensitivity to temporal dependencies but does not assess computational overhead and resource requirements. To provide a more comprehensive evaluation, it is recommended to include specific metrics for computational cost, such as training time, inference time, memory usage, and FLOPs. By reporting these metrics, the paper could offer a clearer picture of the model’s practical feasibility, particularly for deployment in large-scale, real-time, or resource-constrained environments.
2.	The experiment in Section 4.3 shows that the model’s performance significantly declines when the temporal order is shuffled, indicating a strong dependency on temporal features. The high dependency may decrease the model’s effectiveness on datasets that don’t have clear temporal patterns, limiting its application domain.

**Questions:**

1.	Given the complexity of TVDN’s dual-phase network structure, could the authors provide a quantitative comparison of computational costs versus accuracy gains relative to baseline models to allow for a more objective assessment of potential trade-offs between accuracy and efficiency?
2.	In real-world applications, temporal sequences are often affected by various types of noise, which may impact model performance. Could the paper conduct experiments by introducing specific noise types, like Gaussian noise, to simulate random disturbances or missing values to simulate incomplete data, to observe TVDN's performance in these noisy environments, and to assess its robustness to noise?
3.	This paper has shown that TVDN performs excellently on various datasets. Is there any limitation to the model? In cases of data scarcity or extremely complex intervariable relationships, can the model maintain its performance?

---

> ### Author Response · Authors · 2024-11-24
>
> We sincerely thank reviewer **Jesy** for their thorough evaluation of our manuscript, recognition of our decoupling approach, and acknowledgment of our experiments' effectiveness and extensiveness. We are also profoundly grateful for the constructive suggestions regarding computational costs, model robustness, and other limitations.
>
> In response to these valuable suggestions, we have conducted a more comprehensive evaluation of the model, including noise resistance, handling missing values, and efficiency analysis. The revisions have been highlighted in the manuscript and marked with **Jesy** for easy reference.

---

> ### Author Response · Authors · 2024-11-24
> **Response to Weakness 1**
>
> **Weakness 1**
> >1. The experiment focuses on predicting accuracy and sensitivity to temporal dependencies but does not assess computational overhead and resource requirements. To provide a more comprehensive evaluation, it is recommended to include specific metrics for computational cost, such as training time, inference time, memory usage, and FLOPs. By reporting these metrics, the paper could offer a clearer picture of the model’s practical feasibility, particularly for deployment in large-scale, real-time, or resource-constrained environments.
>
>
> **Response**
>
> We appreciate the reviewer's suggestion regarding computational metrics. We have conducted comprehensive evaluations of computational overhead and resource requirements. As shown in Table 6 and Figure 8 (Section 4.4), **our TVDN model achieves an excellent balance between performance and efficiency**. These metrics demonstrate that TVDN achieves superior prediction accuracy while maintaining high computational efficiency, making it well-suited for real-world deployment in resource-constrained environments.
>
> *Table: Comprehensive performance comparison of various time series forecasting models(input length=96). The comparison metrics include model parameters (Param), computational complexity*
> (FLOPs), inference time (Time), memory consumption (Memory), and prediction accuracy
> (MSE).
> | Model        | FLOPs (G) | Param (M) | Inference Time (s) | Peak Memory (MB) | MSE   |
> | ------------ | --------- | --------- | ------------------ | ---------------- | ----- |
> | TVDN (ours)  | 0.46      | 1.44      | 0.0020             | 50.25            | 0.186 |
> | iTransformer | 1.67      | 5.15      | 0.0019             | 62.06            | 0.225 |
> | Client       | 0.32      | 1.01      | 0.0016             | 46.33            | 0.209 |
> | DLinear      | 0.04      | 0.14      | 0.0003             | 42.94            | 0.245 |
> | TimesNet     | 612.79    | 150.37    | 0.0625             | 724.97           | 0.220 |
> | FEDformer    | 4.41      | 12.14     | 0.0298             | 246.33           | 0.246 |
> | ETSformer    | 0.85      | 6.57      | 0.0055             | 80.64            | 0.233 |
> | LightTS      | 0.10      | 0.33      | 0.0009             | 43.65            | 0.265 |
> | Autoformer   | 4.41      | 12.14     | 0.0107             | 221.52           | 0.254 |
> | Pyraformer   | 1.21      | 362.29    | 0.0039             | 1434.35          | 0.376 |
> | Informer     | 3.94      | 12.45     | 0.0055             | 218.42           | 0.373 |
> | PatchTST     | 25.73     | 10.74     | 0.0036             | 257.58           | 0.246 |

---

> ### Author Response · Authors · 2024-11-24
> **Response to Weakness 2**
>
> **Weakness 2**
> >The experiment in Section 4.3 shows that the model’s performance significantly declines when the temporal order is shuffled, indicating a strong dependency on temporal features. The high dependency may decrease the model’s effectiveness on datasets that don’t have clear temporal patterns, limiting its application domain.
>
> **Response**
>
> Thank you very much, Reviewer Jesy, for bringing up this important point.
>
> **Section 4.3**: The model's performance presented in Section 4.3 is relative. Even when the temporal order is disrupted, the absolute performance of TVDN remains superior to that of other comparison models, as illustrated in the updated figure in Section 4.3.
>
> As shown in **Figure 6** and the table below, *MSE Before* represents the error before disrupting the temporal information, while *MSE After* represents the error after the disruption. The results indicate that TVDN exhibits the largest relative changes in MSE and MAE, suggesting that TVDN captures more temporal information than other models. Moreover, even after the temporal information is disrupted, TVDN still outperforms other models in terms of MSE and MAE, demonstrating that the negative effects of temporal dependency disruption do not propagate to cross-variable learning.
>
>
> This question highlights one of the reasons we decoupled inter-variable dependency and temporal dependency: to reduce the impact of cross-temporal learning on cross-variable learning, thereby stabilizing the model. Decoupling ensures that datasets with "unclear temporal patterns" only affect the cross-temporal learning stage and not the entire process, reducing the overall impact.
>
> On datasets without clear temporal patterns, the second stage of our model essentially acts as "incremental learning," where the cross-temporal learning stage does not harm the model’s ability to capture inter-variable relationships. The impact is limited to the accuracy improvement brought by cross-temporal learning, which may be smaller in these scenarios.
>
> **Table :** Model performance before and after temporal order disruption. *MSE Before* and *MAE Before* represent the model’s performance metrics prior to the disruption of temporal information, while *MSE After* and *MAE After* represent the metrics post-disruption. The relative change percentages indicate the sensitivity of each model to temporal order disruption. Results highlight that TVDN demonstrates the largest relative changes in MSE and MAE, suggesting its stronger ability to capture temporal dependencies. Despite the disruption, TVDN maintains superior performance compared to other models, indicating that the negative effects of temporal dependency disruption are confined and do not affect inter-variable learning.
>
>
> | Model        | MSE Before | MSE After | MSE Relative Change (%) | MAE Before | MAE After | MAE Relative Change (%) |
> | ------------ | ---------- | --------- | ----------------------- | ---------- | --------- | ----------------------- |
> | Client       | 0.141      | 0.141     | 0.000                   | 0.236      | 0.237     | 0.424                   |
> | iTransformer | 0.148      | 0.148     | 0.000                   | 0.240      | 0.241     | 0.417                   |
> | DLinear      | 0.197      | 0.196     | -0.508                  | 0.282      | 0.279     | -1.064                  |
> | Informer     | 0.312      | 0.319     | 2.244                   | 0.393      | 0.402     | 2.290                   |
> | AutoFormer   | 0.201      | 0.199     | -0.995                  | 0.317      | 0.312     | -1.577                  |
> | TVDN         | 0.132      | 0.140     | 6.061                   | 0.226      | 0.236     | 4.425                   |

---

> ### Author Response · Authors · 2024-11-24
> **Response to  Questions 1**
>
> **Questions 1**
> >Given the complexity of TVDN’s dual-phase network structure, could the authors provide a quantitative comparison of computational costs versus accuracy gains relative to baseline models to allow for a more objective assessment of potential trade-offs between accuracy and efficiency?
>
> **Response**
>
> Thank you very much for your thoughtful question. As noted in our response to **Weaknesses 1**, we have evaluated the model’s accuracy and efficiency. And our TVDN model achieves an excellent balance between performance and efficiency.
>
> **Detalis**: Figure 8 and Table 6 demonstrate that TVDN achieves superior prediction performance with high efficiency. It requires only 0.46G FLOPs, 1.44M parameters, and 50.25MB peak memory, significantly reducing computational and memory overhead compared to models like iTransformer (1.67G FLOPs, 5.15M parameters) and TimesNet (724.97MB). TVDN's inference speed (0.0020s per sample) is comparable to lightweight models like Client (0.0016s) and much faster than TimesNet (0.0625s). Although DLinear has lower costs (0.04G FLOPs, 0.14M parameters), it performs worse in prediction accuracy (MSE: 0.245). These results confirm TVDN's balance between efficiency and accuracy.

---

> ### Author Response · Authors · 2024-11-24
> **Response to Questions 2**
>
> **Questions 2**
> >In real-world applications, temporal sequences are often affected by various types of noise, which may impact model performance. Could the paper conduct experiments by introducing specific noise types, like Gaussian noise, to simulate random disturbances or missing values to simulate incomplete data, to observe TVDN's performance in these noisy environments, and to assess its robustness to noise?
>
> **Response**
>
> Thank you for the valuable feedback. In response to your question, we have evaluated the performance of the TVDN model in noisy environments to assess its robustness. We have added them in section 4.4 and the appendix with highlighted.
>
> **(1) Noise**
>
> We tested the model at different noise levels and the results are shown in **section4.4**. The performance of TVDN decreases as the noise level increases, but the decrease is small and stable, which indicates that it is more resistant to noise and has good performance at different noise levels.
>
> ***Table** The robustness tests of models on the ECL dataset include performance under varying levels of Gaussian noise. The Gaussian noise level \( \sigma \) indicates that 68\% of the noise falls within \( \pm\sigma \) of the standardized data.*
>
>
> | Noise Level | MSE-96 | MAE-96 | MSE-192 | MAE-192 | MSE-336 | MAE-336 | MSE-720 | MAE-720 |
> | ----------- | ------ | ------ | ------- | ------- | ------- | ------- | ------- | ------- |
> | 0.0         | 0.132  | 0.226  | 0.153   | 0.250   | 0.164   | 0.264   | 0.186   | 0.284   |
> | 0.1         | 0.136  | 0.232  | 0.153   | 0.253   | 0.167   | 0.267   | 0.195   | 0.290   |
> | 0.2         | 0.136  | 0.237  | 0.156   | 0.258   | 0.171   | 0.276   | 0.186   | 0.286   |
> | 0.3         | 0.140  | 0.244  | 0.157   | 0.262   | 0.168   | 0.276   | 0.196   | 0.298   |
> | 0.4         | 0.144  | 0.250  | 0.160   | 0.268   | 0.173   | 0.283   | 0.192   | 0.295   |
> | 0.5         | 0.147  | 0.255  | 0.164   | 0.273   | 0.173   | 0.282   | 0.193   | 0.300   |
> | 0.7         | 0.155  | 0.266  | 0.171   | 0.283   | 0.179   | 0.291   | 0.197   | 0.306   |
> | 1.0         | 0.165  | 0.281  | 0.184   | 0.297   | 0.191   | 0.306   | 0.209   | 0.319   |
>
> **(2) Missing Values**
>
> We also tested the model with missing values and the results are shown in **section 4.4**. The performance of TVDN decreases as the missing rate increases, but the decrease is small and stable, which indicates that it is more resistant to missing values.
>
> **Table:** The robustness tests of models on the ECL dataset include performance under different missing rate levels. The missing ratio $m$ indicates that $(m \times 100)$% of the input data points are randomly masked as missing values (set to zero).
>
> | Prediction Length | Metric | Missing Rate = 0.0 | Missing Rate = 0.1 | Missing Rate = 0.3 | Missing Rate = 0.5 | Missing Rate = 0.7 |
> | ----------------- | ------ | ------------------ | ------------------ | ------------------ | ------------------ | ------------------ |
> | 96                | MSE    | 0.132              | 0.140              | 0.147              | 0.156              | 0.168              |
> |                   | MAE    | 0.226              | 0.241              | 0.252              | 0.263              | 0.275              |
> | 192               | MSE    | 0.153              | 0.152              | 0.159              | 0.169              | 0.182              |
> |                   | MAE    | 0.250              | 0.256              | 0.264              | 0.276              | 0.287              |
> | 336               | MSE    | 0.164              | 0.165              | 0.172              | 0.179              | 0.194              |
> |                   | MAE    | 0.264              | 0.271              | 0.280              | 0.289              | 0.301              |
> | 720               | MSE    | 0.186              | 0.185              | 0.191              | 0.198              | 0.216              |
> |                   | MAE    | 0.284              | 0.288              | 0.301              | 0.307              | 0.321              |

---

> ### Author Response · Authors · 2024-11-24
> **Response to Questions 3**
>
> **Questions 3**
> >This paper has shown that TVDN performs excellently on various datasets. Is there any limitation to the model? In cases of data scarcity or extremely complex intervariable relationships, can the model maintain its performance?
>
>
> **Response**
>
> **A. Complexity of Variables**
>
> As shown in the table, the datasets we tested encompass varying numbers of variables, with the Electricity and Traffic datasets exhibiting the most complex inter-variable relationships (with 862 and 321 variables, respectively). TVDN shows relatively greater advantages on these two datasets, indicating that TVDN is proficient in handling scenarios with complex dependencies among variables.
>
> ***Table**: Detailed dataset descriptions. *Dimension* denotes the variate number of each dataset. *Dataset Size* denotes the total number of time points in (Train, Validation, Test) split respectively. *Prediction Length* denotes the future time points to be predicted and four prediction settings are included in each dataset. *Frequency* denotes the sampling interval of time points.*
>
> | Dataset | Dimension | Prediction Length | Dataset Size | Frequency |
> |----------|-----------|-----------------|--------------|-----------|
> | ETTh1, ETTh2 | 7 | {96, 192, 336, 720} | (8545, 2881, 2881) | Hourly |
> | ETTm1, ETTm2 | 7 | {96, 192, 336, 720} | (34465, 11521, 11521) | 15min |
> | Exchange | 8 | {96, 192, 336, 720} | (5120, 665, 1422) | Daily |
> | Weather | 21 | {96, 192, 336, 720} | (36792, 5271, 10540) | 10min |
> | ECL | 321 | {96, 192, 336, 720} | (18317, 2633, 5261) | Hourly |
> | Traffic | 862 | {96, 192, 336, 720} | (12185, 1757, 3509) | Hourly |
>
> **B. Data Scarcity**
>
> - As mentioned in Q2, under extreme data scarcity conditions (with a missing rate of 0.7), the model's performance declines but still maintains high accuracy with a stable decrease.
>
> - We believe that TVDN has a relative advantage in scenarios with data scarcity because we utilize a smaller look-back window, enabling higher accuracy with limited data. In contrast, even when PathTST and DLinear increase their look-back window to 720, their performance is still inferior to that of TVDN (as show in Appendix).
>
>
> **C. Other Limitations**
>
> - When TVDN increases the look-back window from 96 to 720, the gain in prediction accuracy is limited, which may indicate a marginal effect in utilizing historical sequences. Although this issue exists in most models, we believe that TVDN should be able to achieve greater gains, which is a topic for further research.
>
> - In multivariate time series, temporal delays often exist between different variables due to lagged causal relationships. For instance, the current humidity level may correlate more strongly with future precipitation rather than current precipitation. While TVDN can partially capture such delays, it lacks explicit modeling of these temporal relationships. We are working towards incorporating these temporal delays into explicit modeling frameworks.

---

### Official Review · Reviewer_tVnS · 2024-11-03

**Soundness:** 3
**Presentation:** 2
**Contribution:** 2
**Rating:** 5
**Confidence:** 4

**Summary:**

In this paper, the authors argue that overemphasizing temporal dependencies can destabilize Transformer-based models, increasing their sensitivity to noise, overfitting, and weakening their ability to capture inter-variable relationships. Therefore, they propose a new Temporal-Variable Decoupling Networks (TVDN) which decouples the modeling of variable dependencies from temporal dependencies to address this challenge.

**Strengths:**

1.	The proposed TVDN decouples the modeling of variable dependencies and temporal dependencies which useful in long-term time series forecasting.
2.	TVDN also separates the modeling of temporal dependencies into historical dependencies and predictive dependencies, with the latter having received relatively little attention in previous works.

**Weaknesses:**

1.	In line 61, the authors said that “linear models and Cross-Variable Transformers do not extract accurate temporal dependencies because they essentially map historical series as unordered sets … ”, I disagree with the assertion that these models treat historical series as unordered sets. for example, the linear models predict future time steps by the equation $\hat{x}_{i+1}=\omega_{1}x_{i-L+1}+…+\omega_{L}x_{i}$, clearly treating historical series as ordered sets.
2.	In equation (5) and (6), the authors define $T^{0}=Z_{h}+ Z_{CVE}$; however, in figure 2, it seems that the input of PSTD is consists only $Z_{CVE}$.
3.	Also in figure 2, I cannot find the step (2) in the method section.
4.	In line 246, $Z_{proj}$ is mentioned without a prior definition, I believe it should be $Z_{h}$.
5.	In line 215, the notation $Z_{CVE} \in {R}^{O \times D}$ is used, yet I cannot find the definition of $D$.
6.	In line 315,“the permutation-invariant models overlook dynamic temporal features ”seems inappropriate.
7.	In line 332, I disagree with the claim that “we identified that the bottleneck of the traditional Transformer model lies in the ineffective utilization of historical sequence information”, because the experimental results of PatchTST have shown that longer historical sequences will lead better performance. In addition, the reason why the transformer model performs poorly is that it over-captures channel dependencies, which can also be seen in the experimental results of PatchTST and DLinear. Furthermore, the poor performance of the Transformer model can be attributed to its tendency to over-capture channel dependencies, as evidenced by the experimental results of both PatchTST and DLinear.
8.	The experiments are unfair, since PatchTST and DLinear perform better with longer historical windows. Although the authors have conducted experiments with $L=144$, I suggest the authors also set $L=336$ and $L=512$, and compared the results with PatchTST and DLinear.
9.	In Figure 9, Transformer performs better than Transformer Decoder only which is insistent with  ”even with a significant reduction in historical information …”in Line 912.

**Questions:**

please see weaknesses.

**Details Of Ethics Concerns:**

I think there is no ethics concerns.

---

> ### Author Response · Authors · 2024-11-24
>
> We sincerely thank Reviewer **tVnS** for their meticulous and professional review of our work, which affirmed our decoupling strategy and the idea of decomposing temporal dependency modeling into historical and predictive dependency. Your corrections to the details of the paper and the questioning and discussion of certain viewpoints have also provided us with valuable insights and guidance.
>
> We have carefully addressed your comments and made corresponding revisions to the paper, with all changes highlighted and marked with "**tVnS**" for clarity.

---

> ### Author Response · Authors · 2024-11-24
> **Response to Weakness 1**
>
> **Weakness 1**
>
> >1. In line 61, the authors said that “linear models and Cross-Variable Transformers do not extract accurate temporal dependencies because they essentially map historical series as unordered sets … ”, I disagree with the assertion that these models treat historical series as unordered sets. for example, the linear models predict future time steps by the equation $\hat{x}_{i+1}=\omega_{1}x_{i-L+1}+…+\omega_{L}x_{i}$, clearly treating historical series as ordered sets.
>
> **Response**
>
> Thank you so much for your meticulous corrections. We sincerely apologize for the lack of rigor in our previous statement. In response, we have made revisions to the Introduction section.
>
> We think a more accurate expression would be: 'some linear models' . More precisely, we want to express a class of models where the training output remains relatively unchanged when permuting the order of historical sequence elements, such as the simple projection layer: $y=w_1x_1+w_2x_2+...+w_Lx_L$.
>
> If the time order of $x_1$ and $x_2$ is swapped, the only difference for the projection layer during training is the exchange of the positions of $w_1$ and $w_2$. So the model does not recognize the difference in the temporal order of the sequence.

---

> ### Author Response · Authors · 2024-11-24
> **Response to Weakness 2, 3,4, and 5.**
>
> Thank you very much for your careful reading and thorough review of our paper. We sincerely apologize for the errors in formulas and figures related to **Weakness 2-5**, which may have resulted from multiple revisions and oversight on our part. We have re-checked the relevant details and deeply regret any inconvenience this may have caused during your reading.
>
> **Weakness 2**
>
> >In equation (5) and (6), the authors define $T^0=Z_h + Z_{CVE}$ ; however, in figure 2, it seems that the input of PSTD is consists only $Z_{CVE}$.
>
> **Response**
>
> Thank you for pointing this out. Equations (5) and (6) are correct. In Figure 2, we initially aimed to present a high-level conceptual view, which led to some loss of precision in the details. To avoid any misunderstanding, we have corrected Figure 2 to accurately reflect the complete input $T^0=Z_h + Z_{CVE}$.
>
> **Weakness 3**
> >Also in figure 2, I cannot find the step (2) in the method section.
>
> **Response**
> Thank you for this careful observation. We apologize for the oversight - step (2) was indeed missing from the method section. We have now added its complete mathematical formulation.
>
> **Weakness 4**
> >In line 246, $Z_{proj}$is mentioned without a prior definition, I believe it should be $Z_h$.
>
> **Response**
>
> Thank you for catching this. You are absolutely correct — $Z_{proj}$ should indeed be $Z_h$. This was a notation inconsistency, and we have now corrected it. We appreciate your careful review.
>
> **Weakness 5**
> >In line 215, the notation is used, yet I cannot find the definition of $D$.
>
> **Response**
>
> Thank you very much for your insightful comments, which helped me realize that my explanation was not clear enough. In the original text, line 206, we mentioned the relevant definition: “where $V$ is a matrix containing $D$ embedded tokens.” However, this might not have been sufficiently clear. In our method, the number of tokens is equal to the number of variables because we divide tokens along the variable dimension. We have revised the statement for greater clarity to: “$D$ is equal to the number of variables, and S is the length of the time series.”

---

> ### Author Response · Authors · 2024-11-24
> **Response to Weakness 6**
>
> **Weakness 6**
> >the permutation-invariant models overlook dynamic temporal features ”seems inappropriate.
>
> **Response**
>
> Thank you for raising this point. We sincerely apologize for the imprecision and overly absolute tone in our expression. What we intended to convey was a perspective or hypothesis: "Both iTransformer and Client use a cross-variable Transformer architecture, ranking just below TVDN. It shows that models ignoring temporal ordering can capture cross-variable relationships more effectively, partly supporting the hypothesis that learning temporal dependencies may interfere with variable dependencies." We have revised the relevant parts of the paper accordingly and highlighted them.

---

> ### Author Response · Authors · 2024-11-24
> **Response to Weakness 7**
>
> **Weakness 7**
> >In line 332, I disagree with the claim that “we identified that the bottleneck of the traditional Transformer model lies in the ineffective utilization of historical sequence information”, because the experimental results of PatchTST have shown that longer historical sequences will lead better performance. In addition, the reason why the transformer model performs poorly is that it over-captures channel dependencies, which can also be seen in the experimental results of PatchTST and DLinear. Furthermore, the poor performance of the Transformer model can be attributed to its tendency to over-capture channel dependencies, as evidenced by the experimental results of both PatchTST and DLinear.
>
> **Response**
>
> Thank you for raising these critical points. I sincerely apologize for my expression, which caused any misunderstanding or appeared overly absolute. We have made revisions to the paper accordingly. Additionally, I would like to engage in a discussion with you regarding the related viewpoints.
>
> (1) Our statement specifically refers to "traditional Transformer" architectures (such as the vanilla Transformer, AutoFormer, and Informer), where increasing the historical sequence length does not significantly improve performance, and masking substantial historical information only marginally impacts performance [2][3][4]. PatchTST demonstrates improved utilization of historical information through **patch partitioning** and **channel-independent operations**. We consider this an enhancement over traditional Transformer architectures and, therefore, did not categorize it under the "traditional Transformer" group. For greater clarity, we have revised our statement to explicitly reference "vanilla Transformer models [4]."
>
> (2) Regarding the "**over-capture of channel dependencies**," while we agree this is an important aspect, we consider it a subset of the broader issue of "**ineffective historical information utilization.**" In this regard, I believe our viewpoints are not necessarily in conflict.
>
> - While the channel independence in PatchTST improves performance, this factor alone may not fully explain the limitations of traditional Transformers. One of the sources of its performance gains could also be the patch partitioning [5]. So it might demonstrate sufficiency but does not necessarily prove necessity.
>
> - Cross-variable dependencies have been shown to be beneficial in many studies, including our own experiments[2][3]. This partially supports the effectiveness of cross-variable learning.
>
> - Some studies suggest that **cross-temporal attention** may lead to convergence issues, resulting in overfitting [1][5]—an observation that aligns with our experimental findings.

---

> ### Author Response · Authors · 2024-11-24
> **Response to Weakness 8**
>
> **Weakness 8**
>
> >The experiments are unfair, since PatchTST and DLinear perform better with longer historical windows. Although the authors have conducted experiments with $L=144$, I suggest the authors also set $L=336$ and $L=512$, and compared the results with PatchTST and DLinear.
>
> Thank you for your valuable suggestion.
>
> First, **we do agree that different models may perform optimally with different historical window sizes**. To address this, we have added further experiments. When using PatchTST, increasing the historical window size to $L=336$ significantly affects computational efficiency (approximately 90,000 seconds per batch). Due to these computational constraints, we directly compare our results with those reported in the PatchTST and Dlinear paper. We conducted experiments with six different historical window sizes (24, 48, 96, 192, 336, 720). As shown in the Figure 10. TVDN has better MSE in smaller history windows and also better with increased history windows than other models, including PatchTST and Dlinear and including long-term(prediction length=720) and short-term forecasting(prediction length=96). In addition, TVDN's performance will also grow as the historical window size increases. We have added a section to the appendix to discuss this results.
>
> Second, We think that smaller historical windows may have relative advantages in scenarios **with lacking of long history data.** Thus, efficiently utilizing small historical windows may be necessary In some scenarios. So we do not consider this an entirely "unfair comparison." Additionally, our choice of historical window sizes aligns with some previous works for consistency.

---

> ### Author Response · Authors · 2024-11-24
> **Response to Weakness 9**
>
> **Weakness 9**
> >In Figure 9, Transformer performs better than Transformer Decoder only which is insistent with ”even with a significant reduction in historical information …”in Line 912.
>
>
> **Response**
>
> Thank you for this observation. I wonder if the word "insistent " you mentioned was a typo and should actually be "inconsistent." We sincerely apologize if our expression was not specific enough, leading to any misunderstanding. We have made the following modifications in the paper to address this issue.
>
> ”While the Transformer with historical information performs marginally better in most cases, the performance difference compared to the Transformer Decoder (without historical information) is insignificant. In certain cases, the Transformer Decoder even surpasses the full Transformer. This partially supports the hypothesis that the Transformer model may not effectively utilize historical information. This observation is consistent with previous findings indicating that some Transformer-based models do not necessarily achieve better performance with an increased historical sequence length [2][3][4]."

---

> ### Author Response · Authors · 2024-11-24
> **References**
>
> [1] Ilbert R, Odonnat A, Feofanov V, et al. Unlocking the potential of transformers in time series forecasting with sharpness-aware minimization and channel-wise attention[J]. arXiv preprint arXiv:2402.10198, 2024.
>
> [2] Liu Y, Hu T, Zhang H, et al. itransformer: Inverted transformers are effective for time series forecasting[J]. arXiv preprint arXiv:2310.06625, 2023.
>
> [3] Gao J, Hu W, Chen Y. Client: Cross-variable linear integrated enhanced transformer for multivariate long-term time series forecasting[J]. arXiv preprint arXiv:2305.18838, 2023
>
> [4] Zeng A, Chen M, Zhang L, et al. Are transformers effective for time series forecasting?[C]//Proceedings of the AAAI conference on artificial intelligence. 2023, 37(9): 11121-11128.
>
> [5] Tang P, Zhang W. PDMLP: Patch-based Decomposed MLP for Long-Term Time Series Forecastin[J]. arXiv preprint arXiv:2405.13575, 2024.

---

> ### Comment · Reviewer_tVnS · 2024-11-27
>
> Thanks for the authors' efforts in addressing my questions and concerns, however, i also have the following questions:
> 1. For the weakness 1, i am sorry than i cannot understand the meaning of the authors.
> 2. I read the revision of this paper, but i do not agree with the statement "This partially supports the hypothesis that the Transformer model may not effectively utilize historical information..." in Line 1132. If so, why PatchTST has superior performance especially with large lookback window $L=336/512$ which is also a Transformer-based model? I think the key of PatchTST and DLinear is channel-independent, and modeling channel-independence improperly will lead to overfitting[1] instead of "This performance gap suggests that the original Transformer architecture's focus on the temporal dimension actually comes at the expense of learning cross-variable relationships".
>
> I will keep my original score and will discuss with other reviewers to decide my final score.
> [1] Qi, Shiyi, et al. "Enhancing Multivariate Time Series Forecasting with Mutual Information-driven Cross-Variable and Temporal Modeling." arXiv preprint arXiv:2403.00869 (2024).

---

> ### Author Response · Authors · 2024-11-28
> **Response to New Queston1**
>
> **New Queston1**
> >For the weakness 1, i am sorry than i cannot understand the meaning of the authors.
>
> **Original Weakness 1**
> >In line 61, the authors said that “linear models and Cross-Variable Transformers do not extract accurate temporal dependencies because they essentially map historical series as unordered sets … ”, I disagree with the assertion that these models treat historical series as unordered sets. for example, the linear models predict future time steps by the equation $\hat{x}*{i+1}=\omega*{1}x_{i-L+1}+…+\omega_{L}x_{i}$, clearly treating historical series as ordered sets.
>
>
> **Response**
>
> We will revise the original term "**linear models** do not extract accurate temporal dependencies because they essentially map historical series as unordered sets" to "**Some linear models** do not extract accurate temporal dependencies because they essentially map historical series as unordered sets" .
>
> Specifically, ”**some linear models**“ aim to describe a class of linear models that map historical series as unordered sets. Simply put, these models produce identical predictions when trained on the original historical sequence ($M_1$) and when trained on a randomly shuffled historical sequence ($M_2$), as demonstrated by Figure 6 with examples like Client and iTransformer. Below, I use a simple linear model as an example:
>
> ## 1. Basic Setup
>
> Given a historical sequence $S$ and its target value $y$:
>
> - Historical sequence: $S = [x_1, x_2, \dots, x_L]$
> - Target value: $y$
>
> ## 2. Two Training Approaches
>
> ### Approach 1: Original Order Training (M1)
>
> - Input sequence: $S_1 = [x_1, x_2, \dots, x_L]$
> - Model prediction: $\hat{y}_1 = w_1x_1 + w_2x_2 + \dots + w_Lx_L$
> - Initial weights: $W_1 = [1, 1, \dots, 1]$
> - Loss function: $L_1 = (y - \hat{y}_1)^2$
>
> ### Approach 2: Shuffled Order Training (M2)
>
> ***(Swapping $x_1$ and $x_2$)***
>
> - Input sequence: $S_2 = [x_2, x_1, \dots, x_L]$
> - Model prediction: $\hat{y}_2 = w_1x_2 + w_2x_1 + \dots + w_Lx_L$
> - Initial weights: $W_2 = [1, 1, \dots, 1]$
> - Loss function: $L_2 = (y - \hat{y}_2)^2$
>
> ## 3. Proving Training Equivalence
>
> ### Step 0 (Initial State)
>
> Both approaches yield identical initial predictions:
>
> - $M1: \hat{y}_1 = 1 \cdot x_1 + 1 \cdot x_2 + \dots + 1 \cdot x_L$
> - $M2: \hat{y}_2 = 1 \cdot x_2 + 1 \cdot x_1 + \dots + 1 \cdot x_L$
>
> Thus, $\hat{y}_1 = \hat{y}_2$ (due to the commutative property of addition).
>
> ### Step 1 (First Gradient Descent)
>
> Gradients for $M1$:
>
> - $\frac{\partial L_1}{\partial w_1} = -2(y - \hat{y}_1)x_1$
> - $\frac{\partial L_1}{\partial w_2} = -2(y - \hat{y}_1)x_2$
>
> Gradients for $M2$:
>
> - $\frac{\partial L_2}{\partial w_1} = -2(y - \hat{y}_2)x_2$
> - $\frac{\partial L_2}{\partial w_2} = -2(y - \hat{y}_2)x_1$
>
> Since $\hat{y}_1 = \hat{y}_2$, it follows that:
>
> - $\frac{\partial L_1}{\partial w_1}$ for $M1$ equals $\frac{\partial L_2}{\partial w_2}$ for $M2$
> - $\frac{\partial L_1}{\partial w_2}$ for $M1$ equals $\frac{\partial L_2}{\partial w_1}$ for $M2$
>
> ### Step $t$ (Any Training Step)
>
> This correspondence holds throughout training:
>
> - $M1$ weights at step $t$: $W_1^t = [w_1^t, w_2^t, \dots, w_L^t]$
> - $M2$ weights at step $t$: $W_2^t = [w_2^t, w_1^t, \dots, w_L^t]$
>
> ## 4. Final Equivalence
>
> Final predictions are identical:
>
> - $M1: \hat{y}_1 = w_1^t \cdot x_1 + w_2^t \cdot x_2 + \dots + w_L^t \cdot x_L$
> - $M2: \hat{y}_2 = w_2^t \cdot x_2 + w_1^t \cdot x_1 + \dots + w_L^t \cdot x_L$
>
> Thus, $\hat{y}_1 = \hat{y}_2$.
>
> As shown above, shuffling the input sequence only leads to corresponding changes in weight ordering, while the model predictions remain identical. Therefore, essentially, we think these linear models treat historical sequences as unordered sets. These models still maintain the same output for the scrambled input sequence.

---

> ### Author Response · Authors · 2024-11-28
> **Response to New Queston 2**
>
> **Queston 2**
> >I read the revision of this paper, but i do not agree with the statement "This partially supports the hypothesis that the Transformer model may not effectively utilize historical information..." in Line 1132. If so, why PatchTST has superior performance especially with large lookback window
>  which is also a Transformer-based model? I think the key of PatchTST and DLinear is channel-independent, and modeling channel-independence improperly will lead to overfitting[1] instead of "This performance gap suggests that the original Transformer architecture's focus on the temporal dimension actually comes at the expense of learning cross-variable relationships".
>
> **Response**
>
> We sincerely thank the reviewers for re-evaluating our revised manuscript, and we deeply apologize for any unclear explanations in the previous version. We will make further modifications and provide additional discussions in the final version of the paper.
>
> Regarding the statement in line 1132: "the Transformer model may not effectively utilize historical information"
> Here, we refer specifically to the channel-dependent cross-temporal Transformer used in our experiments (referred to as "vanilla Transformer" below). Therefore, we agree that PatchTST effectively utilizes historical information because it employs the patch strategy and channel independence, which distinguishes it from the vanilla Transformer.
>
> Clarification on the statement: "This performance gap suggests that the original Transformer architecture's focus on the temporal dimension actually comes at the expense of learning cross-variable relationships."
>
> First, when we say 'focus on the temporal dimension', we are referring to the approach in vanilla Transformer where tokens are subjectively divided according to time steps. If we were not focusing on the temporal dimension, there would be no inherent reason to divide tokens by time steps and use attention mechanism. We are sorry that this explanation may not have been sufficiently clear in the original text, and we will revise this in the final manuscript.
>
> Second, We strongly agree that improper modeling of channel dependencies can lead to overfitting and that PatchTST mitigates this issue by adopting a channel-independent strategy and compensating for information loss by extending the input sequence. However, this perspective does not conflict with ours. To better illustrate this point, we describe the following logical chain:
>
> -------
> **Channel-dependent Cross-tempteral Transformer ( vanilla Transformer)**
>
> ↓ (a). Problem Generation)
>
> **Modeling Channel Dependencies improperly** /**(Or other Issues)**
>
> ↓ (b). (Overfitting)
>
> **Solution 1: PatchTST | Solution 2: CDAM [1]**
>
> ------
>
> While PatchTST avoids **issue (a)**, it does not explicitly explain its root cause—"Why does the channel-dependent cross-temporal Transformer (vanilla Transformer) improperly model channel dependencies?"
>
> We stated:
> "This performance gap suggests that the original Transformer architecture's focus on the temporal dimension actually comes at the expense of learning cross-variable relationships." Because If the vanilla Transformer correctly extracts cross-variable relationships, its performance should be at least comparable to that of the  channel-dependent cross-variable Transformer. However, experimental results show that the vanilla Transformer performs significantly worse.
>
> **Our intention is to further **explore (a)**, which represents the upper-level cause of (b)**
>
> Some studies suggest that **cross-temporal attention mechanisms** may be a key factor contributing to issue (a) [3].
> Other studies argue that the receptive field for each time step in the vanilla Transformer is too small, which is mitigated by the larger receptive field provided by patches [2].
> **Although there is no definitive conclusion yet, we believe that from the perspective of gradient descent, the vanilla Transformer may converge to wrong solutions during training, leading to overfitting [3].**
> Therefore, we propose that using a cross-variable Transformer in the first stage could provide a better initialization for optimization, thereby reducing the probability of falling into wrong solutions during the second stage of cross-temporal learning.
>
> **References**
>
> [1] Qi, Shiyi, et al. "Enhancing Multivariate Time Series Forecasting with Mutual Information-driven Cross-Variable and Temporal Modeling." arXiv preprint arXiv:2403.00869 (2024).
>
> [2] Liu Y, Hu T, Zhang H, et al. itransformer: Inverted transformers are effective for time series forecasting[J]. arXiv preprint arXiv:2310.06625, 2023.
>
> [3]Ilbert R, Odonnat A, Feofanov V, et al. Unlocking the potential of transformers in time series forecasting with sharpness-aware minimization and channel-wise attention[J]. arXiv preprint arXiv:2402.10198, 2024.

---

> > ### Comment · Reviewer_tVnS · 2024-12-02
> >
> > Thanks for the author's response. I am curious why you train the model with shuffled example? In my mind, if you train the model with $X_1=[x_1,...,x_n]$, and $y_1$, and then test the model with $x_2=[x_2,x_1,...,x_n]$,  it is  normal that the new output $y_2 \neq  y_1$. On the other hand, if you train a model with $X_2 \neq X_1$ and $y_1 \neq y_2$, it is also normal that $\hat{y}_1=\hat{y}_2$.  I think cannot fully understand your meaning, please give me a more detailed explanation.

---

> > > ### Author Response · Authors · 2024-12-03
> > >
> > > **Comment**
> > > >Thanks for the author's response. I am curious why you train the model with shuffled example? In my mind, if you train the model with $X_1=[x_1,\dots,x_n]$ and $y_1$, and then test the model with $x_2=[x_2,x_1,\dots,x_n]$, it is normal that the new output $y_2\neq y_1$.
> > > On the other hand, if you train a model with $X_2\neq X_1$ and $y_1\neq y_2$, it is also normal that $\hat{y}_1=\hat{y}_2$. I think cannot fully understand your meaning, please give me a more detailed explanation.
> > >
> > >
> > > **Response**
> > >
> > > Thank you for your careful review and response.
> > >
> > > We train the model with shuffled examples and test with the same shuffled order to examine whether a model truly learns the sequential order.
> > >
> > > When testing, we follow the same shuffling order as in training, therefore:
> > >
> > > - If we train with $X_1=[x_1, ..., x_n]$, we test with $X_2=[x_1, ..., x_n]$, output is $y_1$
> > > - If we train with $X_1=[x_2, ..., x_n]$, we test with $X_2=[x_2, ..., x_n]$ using the same shuffling order, output is $y_2$
> > >
> > > So this argues against the statement "it is normal that the new output $y_1\neq y_2$", because in the linear model I mentioned before, $y_1 = y_2$ (Proof in "Response to New Queston 2").
> > >
> > > Let me use a simple 1D CNN comparison to illustrate this point. In 1D CNN model, $y_1 \neq y_2$, which means it is influenced by temporal order. In fact, since fully connected layers treat all time steps with "equal weights" before training, the optimization direction will not change with the input sequence order. However, 1D CNN or Transformers with Position Embedding incorporate positional information, making them sequence-sensitive.
> > > We aim to differentiate these two types of models because simple projection layers or iTransformer show unchanged performance after sequence shuffling, while our TVDN's performance is affected by temporal order due to its sequence sensitivity. However, thanks to the decoupling strategy, this performance degradation does not spread its negative impact to cross-variable learning, still outperforming iTransformer. The two types of models are distinguished as follows:
> > >
> > > ## For Linear Model
> > >
> > > - M_1: Train with $X_1=[x_1,...,x_n]$, test with $X_2=[x_1, x_2, .., x_n]$ to get $y_1$
> > > - M_2: Train with shuffled $X_1=[x_1,x_3, x_2,...,x_n]$, test with same shuffled order $X_2=[x_1,x_3, x_2,...,x_n]$ to get $y_2$
> > > - $y_1=y_2$ indicates that Linear Model hasn't learned temporal order(Proof in "Response to New Queston 2")
> > >
> > > ## For 1D CNN
> > >
> > > - M_1: Train with $X_1=[x_1,...,x_n]$, test with $X_2=[x_1, x_2, .., x_n]$ to get $y_1$
> > > - M_2: Train with shuffled $X_1=[x_1,x_3, x_2,...,x_n]$, test with same shuffled order $X_2=[x_1,x_3, x_2,...,x_n]$ to get $y_2$
> > > - $y_1\neq y_2$ shows that 1D CNN is influenced by temporal order
> > >
> > >
> > > **Proof of 1D CNN's Sequence Order Sensitivity**
> > > 1. Basic Setup
> > > Given a sequence $S$ and its target value $y$:
> > >
> > > - Sequence: $S = [x_1, x_2, x_3]$ (simplified 3-element sequence)
> > > - Target value: $y$
> > > - Convolution kernel size: 2
> > > - Model: 1D CNN with stride 1
> > >
> > > 2. Two Training Approaches
> > >
> > >  Approach 1: Original Order Training (M1)
> > >
> > > - Input sequence: $S_1 = [x_1, x_2, x_3]$
> > > - Model prediction: $\hat{y}_1 = f([w_1x_1 + w_2x_2, w_1x_2 + w_2x_3])$
> > > - Initial weights: $W_1 = [1, 1]$
> > > - Loss function: $L_1 = (y - \hat{y}_1)^2$
> > >
> > > Approach 2: Shuffled Order Training (M2)
> > > (Swapping $x_2$ and $x_3$)
> > >
> > > - Input sequence: $S_2 = [x_1, x_3, x_2]$
> > > - Model prediction: $\hat{y}_2 = f([w_1x_1 + w_2x_3, w_1x_3 + w_2x_2])$
> > > - Initial weights: $W_2 = [1, 1]$
> > > - Loss function: $L_2 = (y - \hat{y}_2)^2$
> > >
> > > 3. Proving Training Inequivalence
> > >
> > > Step 0 (Initial State)
> > >
> > > Initial predictions are different:
> > >
> > > - M1: $\hat{y}_1 = f([1 \cdot x_1 + 1 \cdot x_2, 1 \cdot x_2 + 1 \cdot x_3])$
> > > - M2: $\hat{y}_2 = f([1 \cdot x_1 + 1 \cdot x_3, 1 \cdot x_3 + 1 \cdot x_2])$
> > > Thus, $\hat{y}_1 \neq \hat{y}_2$ (due to positional nature of convolution operation)
> > >
> > > Step 1 (First Gradient Descent)
> > > Gradients for M1:
> > >
> > > $\frac{\partial L_1}{\partial w_1} = -2(y - \hat{y}_1)(x_1 + x_2)$
> > > $\frac{\partial L_1}{\partial w_2} = -2(y - \hat{y}_1)(x_2 + x_3)$
> > >
> > > Gradients for M2:
> > >
> > > $\frac{\partial L_2}{\partial w_1} = -2(y - \hat{y}_2)(x_1 + x_3)$
> > > $\frac{\partial L_2}{\partial w_2} = -2(y - \hat{y}_2)(x_3 + x_2)$
> > >
> > > Since $\hat{y}_1 \neq \hat{y}_2$, gradients are different for M1 and M2.
> > > Step t (Any Training Step)
> > > The difference persists throughout training:
> > >
> > > M1 weights at step t: $W_1^t = [w_1^t, w_2^t]$
> > > M2 weights at step t: $W_2^t = [w_1'^t, w_2'^t]$
> > > where $w_i^t \neq w_i'^t$
> > >
> > > 4. Final Inequivalence
> > > Final predictions are different:
> > >
> > > - M1: $\hat{y}_1 = f([w_1^tx_1 + w_2^tx_2, w_1^tx_2 + w_2^tx_3])$
> > > - M2: $\hat{y}_2 = f([w_1'^tx_1 + w_2'^tx_3, w_1'^tx_3 + w_2'^tx_2])$
> > > Thus, $\hat{y}_1 \neq \hat{y}_2$
> > >
> > > This shows that in 1D CNN, sequence order matters because convolution operations are position-sensitive, unlike linear models where weights and inputs have position-independent relationships.

---

> > > > ### Comment · Reviewer_tVnS · 2024-12-03
> > > >
> > > > Thank you very much, I have read the explanation carefully. I think the core is how you define the "temporal dependencies"? In my mind, the "temporal dependencies" or "ordered set" means that different inputs can lead to different outputs. For example, $y_1=f_{\theta}([x_1,x_2,x_3])$, $y_2=f_{\theta}([x_1,x_3,x_2])$, and $y_1 \neq y_2$ is enough to prove that the model $f_{\theta}$ is sensitive to the order of the input sequence.

---

> > > ### Author Response · Authors · 2024-12-03
> > > **Proof of Linear Model's Sequence Order Independence**
> > >
> > > ## 1. Basic Setup
> > > Given a sequence $S$ and its target value $y$:
> > > - Sequence: $S = [x_1, x_2, x_3]$
> > > - Target value: $y$
> > >
> > > ## 2. Two Training Approaches
> > >
> > > ### Approach 1: Original Order Training (M1)
> > > - Input sequence: $S_1 = [x_1, x_2, x_3]$
> > > - Model prediction: $\hat{y}_1 = w_1x_1 + w_2x_2 + w_3x_3$
> > > - Initial weights: $W_1 = [1, 1, 1]$
> > > - Loss function: $L_1 = (y - \hat{y}_1)^2$
> > >
> > > ### Approach 2: Shuffled Order Training (M2)
> > > ***(Swapping $x_1$ and $x_2$)***
> > > - Input sequence: $S_2 = [x_2, x_1, x_3]$
> > > - Model prediction: $\hat{y}_2 = w_1x_2 + w_2x_1 + w_3x_3$
> > > - Initial weights: $W_2 = [1, 1, 1]$
> > > - Loss function: $L_2 = (y - \hat{y}_2)^2$
> > >
> > > ## 3. Proving Training Equivalence
> > >
> > > ### Step 0 (Initial State)
> > > Both approaches yield identical initial predictions:
> > > - M1: $\hat{y}_1 = 1 \cdot x_1 + 1 \cdot x_2 + 1 \cdot x_3$
> > > - M2: $\hat{y}_2 = 1 \cdot x_2 + 1 \cdot x_1 + 1 \cdot x_3$
> > > Thus, $\hat{y}_1 = \hat{y}_2$ (due to the commutative property of addition)
> > >
> > > ### Step 1 (First Gradient Descent)
> > > Gradients for M1:
> > > - $\frac{\partial L_1}{\partial w_1} = -2(y - \hat{y}_1)x_1$
> > > - $\frac{\partial L_1}{\partial w_2} = -2(y - \hat{y}_1)x_2$
> > > - $\frac{\partial L_1}{\partial w_3} = -2(y - \hat{y}_1)x_3$
> > >
> > > Gradients for M2:
> > > - $\frac{\partial L_2}{\partial w_1} = -2(y - \hat{y}_2)x_2$
> > > - $\frac{\partial L_2}{\partial w_2} = -2(y - \hat{y}_2)x_1$
> > > - $\frac{\partial L_2}{\partial w_3} = -2(y - \hat{y}_2)x_3$
> > >
> > > Since $\hat{y}_1 = \hat{y}_2$, it follows that:
> > > - $\frac{\partial L_1}{\partial w_1}$ for M1 equals $\frac{\partial L_2}{\partial w_2}$ for M2
> > > - $\frac{\partial L_1}{\partial w_2}$ for M1 equals $\frac{\partial L_2}{\partial w_1}$ for M2
> > >
> > > ### Step t (Any Training Step)
> > > This correspondence holds throughout training:
> > > - M1 weights at step t: $W_1^t = [w_1^t, w_2^t, w_3^t]$
> > > - M2 weights at step t: $W_2^t = [w_2^t, w_1^t, w_3^t]$
> > >
> > > ## 4. Final Equivalence
> > > Final predictions are identical:
> > > - M1: $\hat{y}_1 = w_1^t \cdot x_1 + w_2^t \cdot x_2 + w_3^t \cdot x_3$
> > > - M2: $\hat{y}_2 = w_2^t \cdot x_2 + w_1^t \cdot x_1 + w_3^t \cdot x_3$
> > > Thus, $\hat{y}_1 = \hat{y}_2$
> > >
> > > This proves that shuffling the input sequence only leads to corresponding changes in weight ordering, while the model predictions remain identical. Therefore, these linear models essentially treat historical sequences as unordered sets.

---

> ### Author Response · Authors · 2024-12-03
>
> **Comment**
> > Thank you very much, I have read the explanation carefully. I think the core is how you define the "temporal dependencies"? In my mind, the "temporal dependencies" or "ordered set" means that different inputs can lead to different outputs. For example, $y_1 = f_\theta([x_1, x_2, x_3])$, $y_2 = f_\theta([x_1, x_3, x_2])$, and $y_1 \neq y_2$ is enough to prove that the model $f_\theta$ is sensitive to the order of the input sequence.
>
> **Response**
>
> Thank you very much for your response. Yes, from your perspective, sensitivity to different input orders producing different outputs is one type of temporal sensitivity. Your understanding represents a broader interpretation.
>
> However, this approach doesn't effectively evaluate whether a model truly extracts temporal order features. We aim to assess from a more fundamental perspective whether a model has genuinely learned these temporal relationships.
>
> **So our definition of temporal dependency refers to how the predictive performance depends on "the sequential order between $X_{t_i}$ and $X_{t_j}$" - that is, the dependency of model performance on which input comes first and which comes later.**
>
> From a time series perspective, order represents causality, which is crucial. This is why we completely shuffled the training data. A shuffled sequence means the temporal information is entirely lost - the model is essentially receiving a completely different time series. Yet some models remain completely unaffected by this reordering.
>
> Therefore, we defined this evaluation method. For any time series model, we can use this testing approach to verify whether it truly considers the temporal order of sequences. This provides a more rigorous way to assess whether a model has learned meaningful temporal dependencies rather than just exhibiting surface-level order sensitivity.
>
> This method helps distinguish between models that are merely responsive to input ordering and those that genuinely learn and leverage temporal relationships.
>
> Thank you again for your reply.

---

### Official Review · Reviewer_hsgf · 2024-11-05

**Soundness:** 3
**Presentation:** 2
**Contribution:** 2
**Rating:** 3
**Confidence:** 4

**Summary:**

In this paper, the authors investigate the problem of multivariate time series forecasting. The authors claim that overemphasizing temporal dependencies can destabilize the model making the model sensitive to noise. And the simultaneous learning of time-related and variable-related patterns can lead to harmful interference. So the authors propose the temporal-variable decoupling network to model the temporal and variable relationships respectively. The authors evaluate the proposed method on several datasets and achieve good performance.

**Strengths:**

N.A.

**Weaknesses:**

1.	The relationship of inter-variables in the time sequence should have an instantaneous effect, but I am surprised that the author did not quote related work in this area [1,2].
2.	Why do learning time-related and variable-related patterns simultaneously harm inference? Please provide some rigorous evidence.
3.	The author said that Transformer puts too much emphasis on extracting time-dependent patterns and did not provide enough evidence that this statement was suspicious of the rationality of this assertion. From the perspective of the data generation process, future data generation has also received the effects of time delay and cross variables at the same time, which is the opposite of separating these two factors at the same time. This method highlights the intra-and inter- time series relationships, but many methods have studied this problem, such as [3].
4.	Why does the proposed method need to model cross-variable relationships and then model cross-temporal relationships? What is the motivation behind it, what if it is in turn?
5.	More recent comparison methods need to be considered, for example [4] [5].

[1] . Identification of hidden sources by estimating instantaneous causality in high-dimensional biomedical time series.
[2] . Modeling nonstationary time series and inferring instantaneous dependency, feedback and causality: An application to human epileptic seizure event data.
[3] Yu, Guoqi, et al. "Revitalizing multivariate time series forecasting: Learnable decomposition with inter-series dependencies and intra-series variations modeling." arXiv preprint arXiv:2402.12694 (2024).
[4] Jia, Yuxin, et al. "WITRAN: Water-wave information transmission and recurrent acceleration network for long-range time series forecasting." Advances in Neural Information Processing Systems 36 (2024).
[5] Xu, Zhijian, Ailing Zeng, and Qiang Xu. "FITS: Modeling time series with $10 k $ parameters." arXiv preprint arXiv:2307.03756 (2023).

**Questions:**

Please refer to Weaknesses.

---

> ### Author Response · Authors · 2024-11-24
> **Response to Weakness 1**
>
> **Weakness 1**
> > The relationship of inter-variables in the time sequence should have an instantaneous effect, but I am surprised that the author did not quote related work in this area [1,2].
>
> **Response**
>
> Thank you for highlighting this important aspect and providing the relevant literature, which has made our paper more comprehensive. We have discussed the references you mentioned in the Introduction. While our paper focuses on developing a general foundation model for various temporal data types, emphasizing the interaction between cross-variable and temporal dependencies, we should have explicitly discussed these temporal relationship types.
>
> Indeed, multivariate time series can exhibit both instantaneous effects and lagged effects. For example, in a biomedical time series, multiple physiological signals (e.g., heart rate and blood pressure) may be transiently correlated simultaneously. In some cases, there may be delayed effects between some variables. For example, the impact of temperature change on plant growth is usually gradual.
>
> You might be concerned that our decoupling strategy does not account for transient effects at the time level in cross-variate learning. A more detailed explanation is provided below, and we have also included a related discussion in the appendix.
>
> - **Cross-variable learning: Can capture interactions between variables at the same and different timesteps but overlook the specific time ordering.** In the cross-variable learning stage, the model can capture interactions between variables at different timesteps ($V^i_{t}$ and $V^j_{(t + \Delta)}$).The temporal offset $\Delta$ allows the model to capture instantaneous effects (when $\Delta = 0$) and lagged effects (when $\Delta \neq 0$). Thus, we can learn interactions between variables at different time steps during the cross-variate learning phase, ignoring the sequence's temporal order.
>
> - **Cross-temporal learning: Incremental learning instead of siloed learning.** Our temporal learning component incrementally builds upon the cross-variable relationships identified in the first stage. Instead of treating these interactions in isolation, we integrate them to capture instantaneous and lagged effects better. This comprehensive approach ensures that our model effectively captures complex temporal dynamics, including direct and delayed influences between variables.

---

> ### Author Response · Authors · 2024-11-24
> **Response to Weakness 2**
>
> **Weakness 2**
> >Why do learning time-related and variable-related patterns simultaneously harm inference? Please provide some rigorous evidence.
>
> **Response**
>
> Thank you for this critical question, which has helped make our paper more rigorous and comprehensive. Your comment made us realize that we were not clear and complete before. We have added more supporting evidence in the Introduction. The "harm inference" mentioned by the reviewer likely refers to the simultaneous learning of time-related and variable-related patterns, which may cause harmful interference between the two and affect the model's performance during the inference phase. Based on this issue, I will address it from the following two perspectives:
>
> **A. Experimental Evidence**
>
> The ablation studies in our paper demonstrate that separating the learning process significantly improves model performance compared to simultaneous learning. We hypothesize that the challenges in temporal learning (discussed in section B) may lead the model to suboptimal local minima, compromising the starting point for cross-variable learning.
>
> **B. Theoretical and Empirical Support**
>
> - Cross-temporal Transformers often struggle to converge to their true solutions, tending to fall into poor local minima. Training Transformer architectures effectively while avoiding overfitting remains an open challenge [1].
>
> - Cross-variable Transformer architectures have shown better performance by mitigating the risk of poor local minima and improving the performance[2][3].
>
> - Previous attempts at simultaneous learning, such as Crossformer, have shown inferior performance compared to cross-variable Transformers [4].

---

> ### Author Response · Authors · 2024-11-24
> **Response to Weakness 3**
>
> **Weakness 3**
> >The author said that Transformer puts too much emphasis on extracting time-dependent patterns and did not provide enough evidence that this statement was suspicious of the rationality of this assertion. From the perspective of the data generation process, future data generation has also received the effects of time delay and cross variables at the same time, which is the opposite of separating these two factors at the same time. This method highlights the intra-and inter- time series relationships, but many methods have studied this problem, such as [3].
>
> **Response**
>
> Thank you for your detailed response. Your comments made me realize that my expression was imprecise and potentially misleading. We have revised and added clarifications in the main text of the paper. I will address each point systematically and provide comprehensive responses.
>
> **A. Regarding "emphasis on extracting time-dependent patterns:**
>
> Thank you for pointing out this issue, which reminds us that our expression might lead to misunderstanding or doubt. Our intention was to convey that the vanilla approach divides time series by time points, emphasizing time-dependence. However, this time-based division results in performance comparable to or even worse than a simple linear baseline [5].
>
> And our experimental results also show that the original cross-temporal Transformer performs poorly. The time-step-based vanilla Transformer shows poor performance and tends to overfit, while the cross-variable Transformer that focuses on inter-variable relationships performs significantly better[2][3]. This performance gap suggests that the original Transformer architecture's focus on the temporal dimension actually comes at the expense of learning cross-variable relationships.
>
> **B. Regarding the concern about simultaneous effects:**
>
> As explained in our response to Weakness 1:
>
> Our approach does not ignore cross-variable interactions across different time steps; we simply do not consider the variable order. Specifically, we consider the joint effect of $V^i_{t}$ and $V^j_{(t + \Delta{t})}$, but do not differentiate the temporal order between them.
>
> Our temporal learning is actually an "incremental learning" process rather than complete "isolation." In the second phase, we supplement temporal influences on top of the cross-variable learning foundation. This allows us to account for both variable interactions and temporal effects simultaneously, which aligns with rather than contradicts the simultaneous effects of time delay and cross variables.
>
> **C. Regarding the novelty of our approach:**
>
> - Previous methods, including reference [3] cited by reviewer hsgf, did not fully decouple cross-variable learning from temporal learning. Our method employs a serial structure where the negative effects of temporal learning do not propagate to cross-variable learning (as shown in Figure 6, even when temporal dependencies are disrupted, our model still outperforms other models).
>
> - Our experimental results demonstrate superior performance compared to reference [3] in your comments.
>
> - Previous approaches have not decomposed temporal dependencies into historical dependencies and predictive dependencies as we have.

---

> ### Author Response · Authors · 2024-11-24
> **Response to Weakness 4**
>
> **Weakness 4**
> >Why does the proposed method need to model cross-variable relationships and then model cross-temporal relationships? What is the motivation behind it, what if it is in turn?
>
> **Response**
>
> Thank you very much for your critical and insightful question. Your comments reminded me to emphasize this issue. We have added a discussion and experimental results on this topic in the appendix. I will address this from both theoretical references and experimental results.
>
> - Theoretical Motivation:
>
> Previous studies have highlighted that Cross-temporal Transformers are prone to bad local minima and are harder to converge to their true solutions [1][2]. So modeling cross-temporal relationships first can provide an unstable optimization starting point for subsequent cross-variable learning. In contrast, starting with cross-variable modeling helps establish a stable inter-variable relationship structure[2][3], which in turn provides a better optimization starting point for cross-temporal learning. This order increases the likelihood of convergence to the true solution and improves the overall performance of the model.
>
> - Experimental Evidence:
>
> To validate the importance of this modeling order, we conducted experiments where the order of learning was reversed. The results clearly demonstrate that the proposed sequence of learning cross-variable relationships first (CVE) followed by cross-temporal relationships (CTE) outperforms the reversed order.
>
> The results are summarized in the table below:
>
> **Table 1: Performance Comparison of Different Learning Orders on ECL Dataset**
> | Pred_len | CVE → CTE (Proposed) |       | CTE → CVE (Reversed) |       |
> |:--------:|:--------------------:|:-----:|:--------------------:|:-----:|
> |          | MSE                  | MAE   | MSE                  | MAE   |
> | 96       | 0.132                | 0.226 | 0.191                | 0.295 |
> | 192      | 0.153                | 0.250 | 0.194                | 0.293 |
> | 336      | 0.164                | 0.264 | 0.194                | 0.294 |
> | 720      | 0.186                | 0.284 | 0.228                | 0.321 |
>
>
> As shown on the table, across all prediction lengths, the proposed CVE → CTE order consistently achieves lower MSE and MAE compared to the reversed order.

---

> ### Author Response · Authors · 2024-11-24
> **Response to Weakness 5**
>
> **Weakness 5**
> >More recent comparison methods need to be considered, for example [4] [5].
>
> **Response**
>
> Thank you very much for your suggestion. We have added the comparison results of [4][5] of your comments in Appendix of our paper. The comparison results show that the proposed method outperforms the existing methods in terms of both MSE and MAE.

---

> ### Author Response · Authors · 2024-11-24
> **References**
>
> [1] Ilbert R, Odonnat A, Feofanov V, et al. Unlocking the potential of transformers in time series forecasting with sharpness-aware minimization and channel-wise attention[J]. arXiv preprint arXiv:2402.10198, 2024.
>
> [2] Liu Y, Hu T, Zhang H, et al. itransformer: Inverted transformers are effective for time series forecasting[J]. arXiv preprint arXiv:2310.06625, 2023.
>
> [3] Gao J, Hu W, Chen Y. Client: Cross-variable linear integrated enhanced transformer for multivariate long-term time series forecasting[J]. arXiv preprint arXiv:2305.18838, 2023
>
> [4] Zhang Y, Yan J. Crossformer: Transformer utilizing cross-dimension dependency for multivariate time series forecasting[C]//The eleventh international conference on learning representations. 2023.
>
> [5] Zeng A, Chen M, Zhang L, et al. Are transformers effective for time series forecasting?[C]//Proceedings of the AAAI conference on artificial intelligence. 2023, 37(9): 11121-11128.

---

> > ### Comment · Reviewer_hsgf · 2024-11-28
> >
> > Thanks for your response. Regarding cross-variable and cross-temporal learning, it is still unclear why the authors need to decouple these two dependencies. Because in time series analysis, many tools such as Granger Causality can consider them together. From the perspective of data generation, these two dependencies occur simultaneously, so they should not be considered separately. In addition, the author emphasizes learning cross-variables and cross-temporal relationships but does not provide evidence and theory to prove that the learned relationships are correct, which cannot be proved by predictive performance alone. The model proposed in this article is more like a stack of neural networks, so it is difficult to convince people. I will keep my score.

---

> > > ### Author Response · Authors · 2024-11-29
> > > **Response to Official Comment by Reviewer hsgf**
> > >
> > > **Comment**
> > > >Thanks for your response. Regarding cross-variable and cross-temporal learning, it is still unclear why the authors need to decouple these two dependencies. Because in time series analysis, many tools such as Granger Causality can consider them together. From the perspective of data generation, these two dependencies occur simultaneously, so they should not be considered separately. In addition, the author emphasizes learning cross-variables and cross-temporal relationships but does not provide evidence and theory to prove that the learned relationships are correct, which cannot be proved by predictive performance alone. The model proposed in this article is more like a stack of neural networks, so it is difficult to convince people. I will keep my score.
> > >
> > >
> > > **Response**
> > >
> > > Thanks for your comment.
> > >
> > >
> > > First, I believe that the statement "these two dependencies occur simultaneously, so they should not be considered separately" may require further discussion. "Granger Causality can consider them together" doesn't mean that we can correctly model both dependencies at the same time by neural networks. Directly combining these dependencies in deep learning models often increases complexity and risks overfitting, particularly in scenarios with distribution drift[1][7].
> > >
> > > In multivariate time series forecasting, there are two types of modeling approaches from the perspective of multivariate modeling methods: channel independent (CI) and channel dependent (CD). CI treats the multivariate series as multiple univariate forecasting tasks, where each variable is modeled independently. CD, on the other hand, models the variables jointly, considering the relationships between them.
> > >
> > > - In the realm of multivariate time series forecasting, many outstanding works such as PatchTST [2] (ICLR 2023), DLinear [3] (ICLR 2023), PDF [4] (ICLR 2024), and SparseTSF (ICML 2024) have adopted the CI approach, directly discarding the relationships between variables.
> > >
> > > - iTransformer (ICLR 2024) ignores temporal dependencies and only considers cross-variable dependencies (as it shows no impact on prediction results even when temporal order is disrupted, as illustrated in Figure 6).
> > >
> > >
> > > These studies collectively counter the claim that "these two dependencies occur simultaneously, so they should not be considered separately."
> > >
> > > In fact, CI and CD represent a trade-off between capacity and robustness[1]. CD models are more complex but may also be more sensitive to distribution drifts [1]. Therefore, we argue that decoupling the two dependencies and allowing each to play to its strengths is a relatively natural idea.
> > >
> > > Second, regarding the criticism that we "not provide evidence and theory to prove that the learned relationships are correct," we believe predictive performance itself is part of the supporting evidence. Moreover, we explicitly design the TVDN model with separate CVE and CTE components, both of which employ interpretable mechanisms such as self-attention and 1D CNNs to explicitly model cross-variable and cross-temporal dependencies. This design is not a random stacking of neural networks. More importantly, our ablation experiments (section 4.2)  demonstrate that CTE learns cross-temporal dependencies while CVE only learns cross-variable dependencies. This is evidenced by the fact that disrupting temporal order does not affect CVE but does affect CTE. By decoupling the two dependencies, CVE remains unaffected by temporal disruption, enabling the model to rely on cross-variable modeling to achieve accurate predictions even under disrupted temporal conditions, thereby enhancing its robustness.
> > >
> > > **References**:
> > > [1] Han L, Ye H J, Zhan D C. The capacity and robustness trade-off: Revisiting the channel independent strategy for multivariate time series forecasting[J]. IEEE Transactions on Knowledge and Data Engineering, 2024.
> > > [2] Nie Y, Nguyen N H, Sinthong P, et al. A time series is worth 64 words: Long-term forecasting with transformers[J]. arXiv preprint arXiv:2211.14730, 2022.
> > > [3] Zeng A, Chen M, Zhang L, et al. Are transformers effective for time series forecasting?[C]//Proceedings of the AAAI conference on artificial intelligence. 2023, 37(9): 11121-11128.
> > > [4] Dai T, Wu B, Liu P, et al. Periodicity decoupling framework for long-term series forecasting[C]//The Twelfth International Conference on Learning Representations. 2024.
> > > [5] Lin S, Lin W, Wu W, et al. SparseTSF: Modeling Long-term Time Series Forecasting with 1k Parameters[J]. arXiv preprint arXiv:2405.00946, 2024.
> > > [6] Liu Y, Hu T, Zhang H, et al. itransformer: Inverted transformers are effective for time series forecasting[J]. arXiv preprint arXiv:2310.06625, 2023.
> > > [7] Ilbert R, Odonnat A, Feofanov V, et al. Unlocking the potential of transformers in time series forecasting with sharpness-aware minimization and channel-wise attention[J]. arXiv preprint arXiv:2402.10198, 2024.

---

### Author Response · Authors · 2024-11-26
**Overall response**

We sincerely thank all reviewers for their constructive comments and suggestions. We have improved the manuscript by incorporating their feedback, especially additional results and discussions on motivation, performance, and efficiency. We have uploaded a revised paper with new material highlighted and marked with reviewer names.

As reviewers  noted, our core contribution is decoupling cross-variable and temporal dependency learning, and integrating historical and prediction sequences in our prediction paradigm. With less overhead, we achieved  SOTA accuracy on multiple datasets, demonstrated robustness under various noise levels and missing rates, and showed improved performance with longer historical windows.

Beyond these, we sincerely hope that the reviewers will consider the inspirational aspects of our work:

- **Complete Decoupling in Gradient Propagation**: Our decoupling is entirely in the gradient propagation dimension. This strategy may have **general applicability** in multivariate time series forecasting. Because as shown in Section 4.2, for all used datasets, models focusing on cross-variable learning show minimal or negative accuracy gains, while **switching to temporal learning leads to continuous improvements.** And combining both modes may result in slight gains or negative growth. This could inspire future work to decouple different patterns at the gradient level (eg., decomposing temporal patterns like seasonality and trends).

- **Integrating Historical and Prediction Sequences**: Dividing time series into historical and prediction sequences differs from methods focusing only on prediction sequences. We construct the prediction sequence $s_{\text{prediction}}$ in the first phase and form a complete sequence $z = [y_{\text{history}}, y_{\text{prediction}}]$. Using $z$ as input for another phase transforms the task from $x \rightarrow y_{\text{prediction}}$ to fine-tuning $z \rightarrow z'$. This offers two benefits:

  - **Expanded Horizon**: Directly predicting longer horizons achieves better accuracy than multi-step autoregressive decoding [1][2]. Including historical sequences may further expand the horizon.

  - **Leveraging Non-Stationary Information**: Normalizing input data removes non-stationary information, potentially limiting models before denormalization [3][4]. Our joint sequence $z = [y_h, y_p]$ allows the model to reconsider non-stationary information globally. Expanding the receptive field to include non-stationary information could be inspirational (e.g., concatenating $y_{\text{history}}$ and $y_{\text{prediction}}$ may capture local dependencies near their junction).

------
**Summary**

1. In response to reviewer **hsgf**'s comments on instantaneous variable relationships, motivation, effects of time delays and cross-variables, modeling order, and comparisons with new methods:

- **Instantaneous Variable Relationships (Appendix I)**

- **Motivation**: Expanded in Introduction.

- **Simultaneous Effects of Time Delays and Cross-Variables**: Discussed in Appendix I.

- **Modeling Order of Cross-Variables and Temporal Dependencies**: See Appendix H.

- **Comparison with New Methods**: Added in Appendix B.


2. To address reviewer **tVnS**'s comments, we corrected paper details, added experiments on extending historical sequence length, and discussed possible improvement sources in PatchTST and Dlinear. Main revisions:

- **Clarity and Rigor**: Improved in Sections 1, 3, and Appendix F.

- **Experiments on Historical Sequence Length**: Added in Appendix D.


3. For reviewer **Jesy**'s comments, we addressed "strong dependency on temporal features," added efficiency and robustness experiments:

- **Temporal Feature Dependency**: Discussed in Figure 6.

- **Efficiency Analysis**: Added in Section 4.4.

- **Robustness Analysis**: Added in Section 4.4.

---

**References**

[1] Garza A, Mergenthaler-Canseco M. TimeGPT-1. *arXiv preprint* arXiv:2310.03589, 2023.

[2] Zeng A, Chen M, Zhang L, et al. Are transformers effective for time series forecasting? In *Proceedings of the AAAI Conference on Artificial Intelligence*, 2023, 37(9): 11121-11128.

[3] Liu Y, Wu H, Wang J, et al. Non-stationary transformers: Exploring the stationarity in time series forecasting. *Advances in Neural Information Processing Systems*, 2022, 35: 9881-9893.

[4] Ma X, Li X, Fang L, et al. U-Mixer: An Unet-Mixer Architecture with Stationarity Correction for Time Series Forecasting. In *Proceedings of the AAAI Conference on Artificial Intelligence*, 2024, 38(13): 14255-14262.

---

### Meta-Review · Area_Chair_YGe1 · 2024-12-20

**Metareview:**

This paper proposes a Temporal-Variable Decoupling Network (TVDN) to address the issue of overemphasizing temporal dependencies in multivariate time series forecasting, which the authors argue can destabilize models and lead to sensitivity to noise. The paper achieves a state-of-the-art performance on several datasets. However, as raised by the reviewer, the justification of claims and the experiences are somewhat limited.

**Additional Comments On Reviewer Discussion:**

Although the paper has some merits, such as decoupling the modeling of variable dependencies from temporal dependencies and achieving state-of-the-art performance on several datasets, the issues raised by the reviews are critical. For instance, the lack of rigorous evidence supporting the claim that simultaneous learning of time-related and variable-related patterns harms inference (hsgf), the possible problematic statement (tVnS), and the need for a more comprehensive evaluation of computational overhead and resource requirements (Jesy). Although the author addresses some issues in responses, the paper still needs a major revision before it can be accepted, especially in addressing the need for rigorous evidence. The decision is reject.

---

### Decision · Program_Chairs · 2025-01-22

Reject